# OPEN

# Genome-wide analyses of 200,453 individuals yield new insights into the causes and consequences of clonal hematopoiesis

Siddhartha P. Kar [1,2,12] ✉, Pedro M. Quiros [3,4,5,12] ✉, Muxin Gu[3,4], Tao Jiang[6], Jonathan Mitchell[7], Ryan Langdon[1,2], Vivek Iyer[4], Clea Barcena[3,4], M. S. Vijayabaskar[3,4], Margarete A. Fabre[3,4,8], Paul Carter[9], Slavé Petrovski[7,10], Stephen Burgess [6,11] and George S. Vassiliou [3,4,8] ✉

Clonal hematopoiesis (CH), the clonal expansion of a blood stem cell and its progeny driven by somatic driver mutations, affects over a third of people, yet remains poorly understood. Here we analyze genetic data from 200,453 UK Biobank participants to map the landscape of inherited predisposition to CH, increasing the number of germline associations with CH in European-ancestry populations from 4 to 14. Genes at new loci implicate DNA damage repair (*PARP1*, *ATM*, *CHEK2*), hematopoietic stem cell migration/homing (*CD164*) and myeloid oncogenesis (*SETBP1*). Several associations were CH-subtype-specific including variants at *TCL1A* and *CD164* that had opposite associations with *DNMT3A*- versus *TET2*-mutant CH, the two most common CH subtypes, proposing key roles for these two loci in CH development. Mendelian randomization analyses showed that smoking and longer leukocyte telomere length are causal risk factors for CH and that genetic predisposition to CH increases risks of myeloproliferative neoplasia, nonhematological malignancies, atrial fibrillation and blood epigenetic ageing.

The pervasive effects of ageing and somatic mutation shape the landscape of human disease in later life[1]. A ubiquitous feature of ageing is the development of somatic mutation-driven clonal expansions in aged tissues[2,3]. In blood, somatic mutations that enhance cellular fitness of individual hematopoietic stem cells (HSCs) and their progeny give rise to the common age-related phenomenon of CH[4–7]. CH becomes increasingly prevalent with age[4–6] and is associated with an increased risk of hematological cancers[4,5,8,9] and some nonhematological conditions[5,10,11]. However, our understanding of the biological basis for these associations remains limited, as does our ability to explain how CH driver mutations promote clonal expansion of mutant HSCs[12]. In fact, whilst CH is defined by its association with somatic mutations, its development is influenced by nonmutation factors[13–16] and by the heritable genome[17,18], in ways that remain poorly understood.

Insights into the causes and consequences of CH are confounded by its intimate relationship with ageing. Moreover, even when robust associations are identified, their causality can be difficult to establish. Here, we perform a comprehensive investigation of the genetic and phenotypic associations of CH in 200,453 UK Biobank (UKB) participants, yielding a step change in our understanding of CH pathogenesis. Our study reveals multiple new germline loci associated with CH, including several that interact with specific CH subtypes; uncovers causal links between CH and diverse pathological states across organ systems; and provides evidence for causal

associations between smoking and telomere length and CH risk, amongst a series of insights.

## Results

**Overall and gene-specific prevalence of CH by age and sex.** To identify individuals with CH, we analyzed blood whole-exome sequencing (WES) data from 200,453 UKB participants of diverse ancestry[19] aged 38–72 yr (Extended Data Fig. 1a–c). We called somatic mutations in 43 CH genes (Supplementary Table 1) and filtered these against a predefined list of CH driver variants (Supplementary Tables 2 and 3). This identified 11,697 mutations (Supplementary Table 4) in 10,924 individuals (UKB prevalence: 5.45%), displaying patterns in line with previous reports[4,5,17] (Fig. 1a and Extended Data Fig. 1d–h). Interestingly, the age-related rise in CH prevalence differed between driver genes (Fig. 1b and Extended Data Fig. 2a–c), for example, *DNMT3A* prevalence rose earlier in life compared with *SF3B1* and *SRSF2*, consistent with what we now know about the lifelong behavior of these CH subtypes[20]. Females and males were similarly affected overall (Extended Data Fig. 2d); however, there were significant gene-level differences between sexes (Fig. 1c), reflecting the sex-specific differences in prevalence of these gene-level frequencies in myeloid malignancies[21].

**Associations between CH and traits prevalent at baseline.** To identify associations between CH and traits or diseases prevalent at the time of enrollment to the UKB, we performed logistic regression

[1]MRC Integrative Epidemiology Unit, University of Bristol, Bristol, UK. [2]Section of Translational Epidemiology, Division of Population Health Sciences, Bristol Medical School, University of Bristol, Bristol, UK. [3]Department of Haematology, Wellcome-MRC Cambridge Stem Cell Institute, Jeffrey Cheah Biomedical Centre, University of Cambridge, Cambridge, UK. [4]Wellcome Sanger Institute, Wellcome Genome Campus, Hinxton, Cambridge, UK. [5]Instituto de Investigación Sanitaria del Principado de Asturias (ISPA), Oviedo, Spain. [6]BHF Cardiovascular Epidemiology Unit, Department of Public Health and Primary Care, Strangeways Research Laboratory, University of Cambridge, Cambridge, UK. [7]Centre for Genomics Research, Discovery Sciences, BioPharmaceuticals R&D, AstraZeneca, Cambridge, UK. [8]Department of Haematology, Cambridge University Hospitals NHS Foundation Trust, Cambridge, UK. [9]Division of Cardiovascular Medicine, Department of Medicine, Cambridge University Hospitals NHS Foundation Trust, Cambridge, UK. [10]Department of Medicine, University of Melbourne, Austin Health, Melbourne, Victoria, Australia. [11]MRC Biostatistics Unit, University of Cambridge, Cambridge, UK. [12]These authors contributed equally: Siddhartha P. Kar, Pedro M. Quiros. ✉e-mail: siddhartha.kar@bristol.ac.uk; pmquiros@ispasturias.es; gsv20@cam.ac.uk

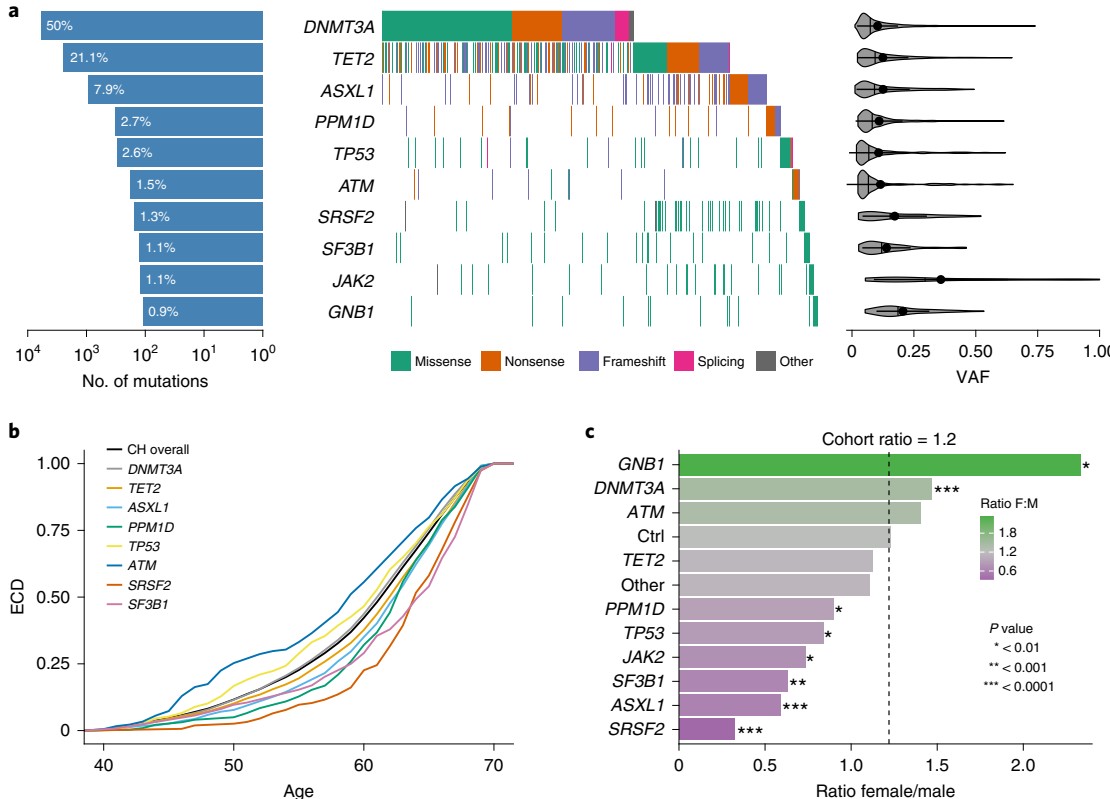

**Fig. 1 | Characterization of CH in the UKB. a**, Composite plot summarizing mutations in the 10 most common driver genes in 10,924 individuals with CH. Each column in the waterfall plot represents a single individual, with mutation types color-coded. Bars on the left quantify mutations per gene as a percentage of all CH mutations identified. Violin plots on the right show the distribution of VAFs, with vertical lines representing the median and dots with horizontal lines the mean ± s.d. **b**, Empirical cumulative distribution (ECD) of the age of individuals with CH overall (black) and stratified by the eight most common driver genes. Compared with *DNMT3A*, mutations in *ATM* were observed 3 yr earlier ($P = 7.2 \times 10^{-4}$), while mutations in *ASXL1*, *PPM1D*, *SRSF2* and *SF3B1* were observed 1 ($P = 2.7 \times 10^{-8}$), 1 ($P = 8.5 \times 10^{-6}$), 2 ($P = 5.7 \times 10^{-10}$) and 3 ($P = 6.5 \times 10^{-6}$) years later, respectively. Differences were calculated using two-sided pairwise Wilcoxon rank sum tests. **c**, Bar plot showing the female to male (F:M) ratio of CH carriers with mutations in the ten most common driver genes. *GNB1* ($P = 2.3 \times 10^{-3}$) and *DNMT3A* ($P = 3.2 \times 10^{-11}$) show a higher F:M ratio, while *PPM1D* ($P = 6.4 \times 10^{-3}$), *TP53* ($P = 1 \times 10^{-3}$), *JAK2* ($P = 5.7 \times 10^{-3}$), *SF3B1* ($P = 3.5 \times 10^{-4}$), *ASXL1* ($P = 5.7 \times 10^{-28}$) and *SRSF2* ($P = 3.8 \times 10^{-14}$) show lower F:M ratio. 'Other' represents the remaining driver genes grouped together and 'Ctrl' the ratio for individuals without CH. Dotted vertical line shows the F:M ratio observed in the full cohort (F:M = 1.2). *P* values are from a chi-squared test comparing the distribution for each gene with 'Ctrl'.

analyses with CH as the outcome in the cohort of 200,453 individuals. We found that age increased the risk of CH by 6.7% per year and that prevalent hypertension, but not obesity or type 2 diabetes (T2D), was associated with CH status (Fig. 2a and Supplementary Table 5). We also found that individuals with CH were more likely to be current or former smokers, an association that held true for different forms of CH and was strongest for *ASXL1*-mutant CH (Fig. 2a and Supplementary Table 5). Analyses of complete blood count and biochemical parameters identified both known and previously unreported associations with overall CH and CH subtypes (Fig. 2a and Supplementary Tables 6 and 7). We also found that CH status was associated with lower prevalent levels of total and low-density lipoprotein cholesterol, most marked for *JAK2* and splicing factor-mutant CH (Fig. 2a and Supplementary Table 7).

**Associations between CH and incident disease.** We next performed a phenome-wide association study (PheWAS) of incident disease in the UKB considering CH at baseline as the exposure. This identified strong associations with myeloid malignancies and associated sequelae (Extended Data Fig. 3a and Supplementary Table 8). Analyses for selected phenotypes (Supplementary Table 9) also identified a high incidence of myeloid malignancies with all forms of CH (Fig. 2b and Supplementary Table 10) and increased risks

of other hematological and nonhematological neoplasia, including lymphoma, lung and kidney cancers (Fig. 2b and Supplementary Table 10). Notably, associations with lung and other cancers were also observed in self-reported never smokers (Extended Data Fig. 3b and Supplementary Table 11). Unlike previous reports linking CH with ischemic cardiovascular disease (CVD)[5,10,22], we did not find a significant association between CH and ischemic CVD, including coronary artery disease (CAD) and stroke; but we did find an association with heart failure and atrial fibrillation, and a composite of all CVD conditions in CH with large clones in multivariable regression models (Fig. 2b and Supplementary Table 10). While CH was associated with CAD and ischemic stroke in unadjusted analyses, adjusting for age led to these associations attenuating to the null, demonstrating the impact of age as a confounder (Extended Data Fig. 3c and Supplementary Table 12). Finally, we also found that CH increased the risk of death from diverse causes (Fig. 2b and Supplementary Table 13).

**Heritability of CH and cell-type-specific enrichment.** To identify heritable determinants of CH risk, we performed a genome-wide association study (GWAS) on the 184,121 individuals with genetically inferred European ancestry to identify common (minor allele frequency (MAF) > 1%) germline genetic variants predisposing to

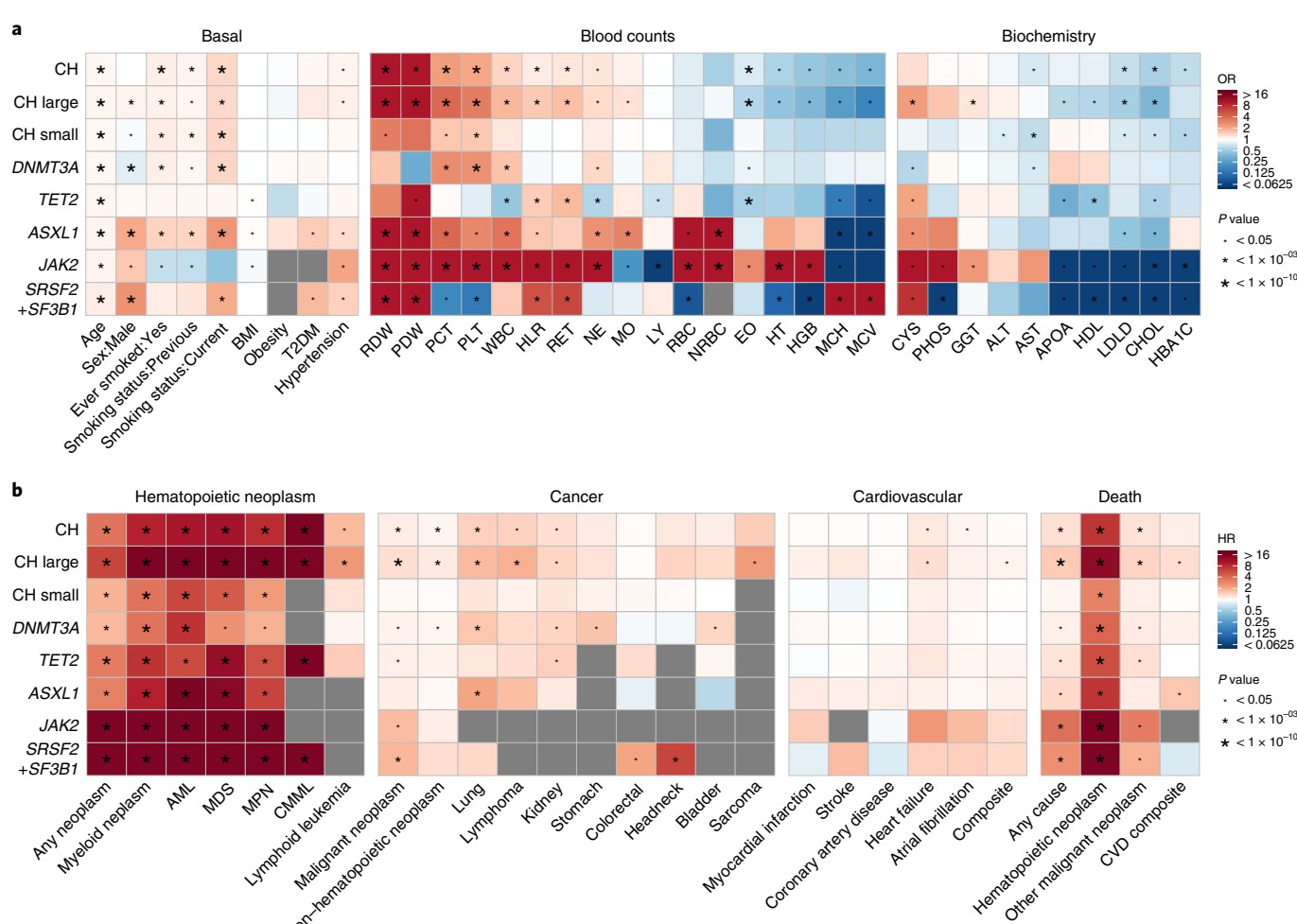

**Fig. 2 | Associations between CH and diverse traits/diseases. a,b,** Heatmaps showing associations between overall CH (CH); CH with large (CH large) and small (CH small) clones; and CH driven by *DNMT3A*, *TET2*, *ASXL1*, *JAK2* and *SRSF2 + SF3B1* mutations, with prevalent (**a**) or incident (**b**) traits/ diseases. Prevalent traits include baseline characteristics, blood counts and serum biochemistry, while incident traits include hematopoietic neoplasms, cancer types, CVDs and death. Red–blue color scale represents the OR or hazard ratio (HR). ORs were calculated using a logistic regression model, while HRs were calculated using competing risk models, except for the death analysis where a Cox proportional hazards model was used. Gray color represents failure of the logistic regression model (maximum likelihood estimation algorithm) to converge. Asterisks represent significant associations, and their size represents different unadjusted *P* value cut-offs. All ORs/HRs, 95% CIs, sample sizes, *P* values and FDRs (to adjust for multiple comparisons) are reported in Supplementary Tables 5–7, 10 and 13. T2DM, type 2 diabetes mellitus; RDW, red blood cell (erythrocyte) distribution width; PDW, platelet distribution width; PCT, plateletcrit; PLT, platelet count: WBC, white blood cell (leukocyte) count; HLR, high light scatter reticulocyte count; RET, reticulocyte count; NE, neutrophil count; MO, monocyte count; LY, lymphocyte count; RBC, red blood cell (erythrocyte) count; NRBC, nucleated RBC; EO, eosinophil count; HT, hematocrit percentage; HGB, hemoglobin concentration; MCH, mean corpuscular hemoglobin; MCV, mean corpuscular volume; CYS, cystatin C; PHOS, phosphate; GGT, gamma-glutamyltransferase; ALT, alanine aminotransferase; AST, aspartate aminotransferase; APOA, apolipoprotein A; HDL, high-density lipoprotein cholesterol; LDLD, low-density lipoprotein direct cholesterol; CHOL, total cholesterol; HBA1C, glycosylated hemoglobin; AML, acute myeloid leukemia; MDS, myelodysplastic syndromes; CMML, chronic myelomonocytic leukemia.

CH. In the GWAS, we compared 10,203 individuals with CH with 173,918 individuals without CH, after quality control (QC) of the germline genotype data. Linkage disequilibrium score regression (LDSC)[23] showed little evidence of inflation in test statistics due to population structure (intercept = 1.009; lambda genomic control factor = 0.999). The narrow-sense (additive) heritability of CH was estimated at 3.57% (s.e. = 0.85%). We partitioned the heritability across four major histone marks observed in 10 cell-type groups aggregated from 220 cell-type-specific annotations[24] and identified strong enrichment of the polygenic CH signal in histone marks enriched in hematopoietic cells (*P* = 5.9 × 10⁻⁵; Fig. 3a and Supplementary Table 14). Next, we partitioned the heritability of CH across open chromatin state regions in various hematopoietic progenitor cells and lineages[24,25]. Previous work on other traits[25,26] has

established that trait heritability tends to be enriched in transcriptionally active open chromatin regions in trait-relevant cell types, helping implicate specific cell types as key mediators of the GWAS signal. Consistent with this, we found CH heritability enrichment in accessible chromatin regions in HSCs, common lymphoid and myeloid progenitors, multipotent and erythroid progenitors, and B cells (Fig. 3b and Supplementary Table 15). Overall, these findings endorse the intuitive assumption that CH associations exert their greatest biological effect on HSC/progenitor populations.

**Germline genetic loci associated with overall CH risk.** Linkage disequilibrium (LD)-based clumping of 10,013,700 common autosomal and X chromosomal variants identified seven independent (*r*² < 0.05) genome-wide significant loci (lead variant

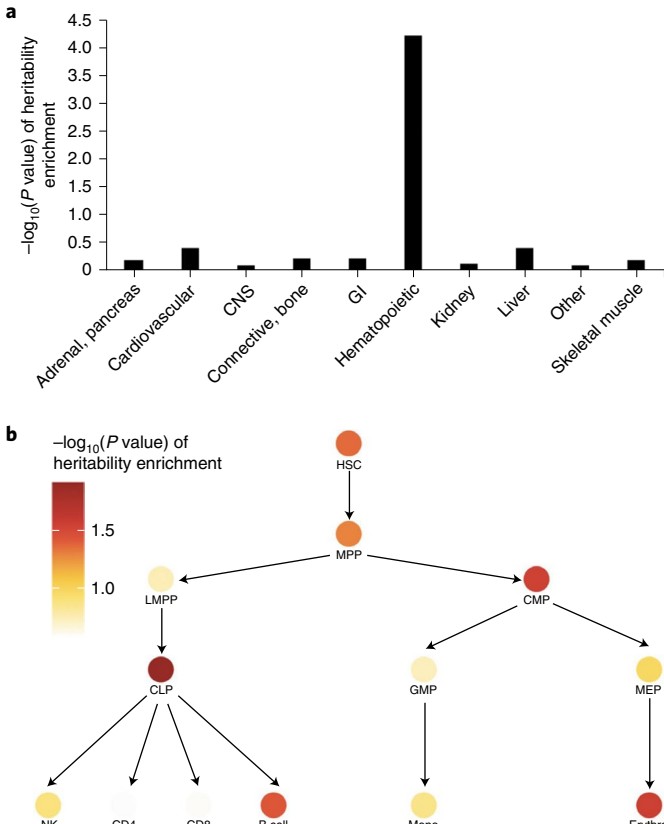

**Fig. 3 | Cell-type-specific enrichment of the CH polygenic signal.**
**a**, Heritability enrichment of CH across histone marks profiled in ten cell-type groups. **b**, Heritability enrichment of CH across open chromatin regions identified by ATAC-seq in hematopoietic progenitor cells/lineages at different stages of differentiation. Partitioned heritability cell-type group analysis in the LDSC software was used to compute these enrichments and corresponding *P* values. The data underlying the figures are available in Supplementary Tables 14 and 15. CNS, central nervous system; GI, gastrointestinal; CLP, common lymphoid progenitor; CMP, common myeloid progenitor; MPP, multipotent progenitor; GMP, granulocyte/macrophage progenitor; LMPP, lymphoid-primed multipotent progenitor; NK, natural killer cell; Mono, monocyte; Erythro, erythroid progenitor.

$P < 5 \times 10^{-8}$) associated with risk of developing CH, including three CH loci previously reported[17] in European-ancestry populations: two at 5p15.33-*TERT* and one at 3q25.33-*SMC4* (Fig. 4a and Supplementary Table 16). We identified a new top variant in the 5p15.33 region, rs2853677 ($P = 2.4 \times 10^{-50}$), which was weakly correlated ($r^2 = 0.19$) with the previously reported[17] top variant, rs7705526 ($P = 3.4 \times 10^{-44}$ in our analysis). Overall, there was evidence for three independent ($r^2 < 0.05$) signals at 5p15.33 marked by lead variants rs2853677, rs13156167 and rs2086132, the latter representing a new signal independent of the two previously published[17] signals, rs7705526 and rs13167280. After approximate conditional analysis[27] (Supplementary Table 17) conditioning on the three lead variants in the *TERT* region, the previously published top variant, rs7705526, continued to remain genome-wide significant, suggesting that it represented a fourth signal in this region. Conditional analysis also highlighted the existence of a fifth independent association at 5p15.33 marked by rs13356700, ~776 kb from *TERT* and ~34 kb from *EXOC3* (Supplementary Table 17), that encodes an exocyst complex component implicated in arterial thrombosis[28]. The variant rs13356700 was in strong LD ($r^2 = 0.84$) with rs10072668 which is associated with hemoglobin

concentration/hematocrit percentage[29]. At 3q25.33-*SMC4*, the previously reported[17] top variant, rs1210060191, was not captured in the UKB and our top association was rs12632224 ($P = 2.3 \times 10^{-9}$). We also identified three other new genome-wide significant loci associated with overall CH susceptibility (Fig. 4a and Supplementary Table 16): 4q35.1-*ENPP6* (rs13130545), 6q21-*CD164* (rs35452836) and 11q22.3-*ATM* (rs11212666). As we were underpowered for GWAS in non-European-ancestry UKB participants, we evaluated associations of the seven lead variants for overall CH in the 505 individuals with CH and 11,893 controls comprising this ancestrally diverse subcohort and found that 6 of 7 variants displayed effect size estimates consistent with those in individuals of European ancestry (Extended Data Fig. 4).

**Stratified CH GWAS and association heterogeneity.** Next, we investigated whether the development of certain CH subtypes is affected by germline variants. Thus, we performed GWAS for four additional CH traits—stratifying by the two main CH genes, *DNMT3A* and *TET2*, and by clonal size, defining small clones as those with variant allele fraction (VAF) < 0.1 and large clones by VAF ≥ 0.1. Focusing on 5,185 individuals with *DNMT3A* and 2,041 with *TET2* mutations and 173,918 controls (individuals of European ancestry without CH), we identified eight and three genome-wide significant loci associated with *DNMT3A*- and *TET2*-mutant CH, respectively (Fig. 4b,c and Supplementary Tables 18 and 19). We replicated the only previously published risk locus associated with *DNMT3A*-CH in European-ancestry populations at 14q32.13-*TCL1A*. The overall CH loci at 5p15.33-*TERT* (signals with lead variants rs2853677, rs13156167 and rs7705526), 3q25.33-*SMC4*, 6q21-*CD164* and 11q22.3-*ATM* were also genome-wide significant for *DNMT3A*-CH. We also found two new loci for *DNMT3A*-CH marked by lead variants rs138994074 at 1q42.12-*PARP1* and rs8088824 at 18q12.3-*SETBP1* (Fig. 4b and Supplementary Table 18). The three *TET2*-CH-associated loci included the lead variant rs2736100 at 5p15.33-*TERT*, which was moderately correlated ($r^2 = 0.44$) with the overall CH lead variant rs2853677 in the same region. The other two risk loci, both new for *TET2*-CH, were at lead variants rs10131341 (14q32.13-*TCL1A*) and rs79633204 (7q32.2-*TMEM209*; Fig. 4c and Supplementary Table 19). Notably, the A allele of rs10131341 had opposite associations with *TET2*-CH (odds ratio (OR) = 1.28, $P = 6.8 \times 10^{-10}$) versus *DNMT3A*-CH (OR = 0.87, $P = 6.4 \times 10^{-8}$). A trend for opposite effects at 14q32.13-*TCL1A* was also observed in a previous study[17], but did not achieve genome-wide significance for *TET2*-CH.

When comparing 4,049 individuals with large or 6,154 individuals with small clones against the 173,918 controls without CH, we found that the overall CH loci at 5p15.33-*TERT* and 3q25.33-*SMC4* were associated at genome-wide significance with large clone CH (Fig. 4d and Supplementary Table 20), while 5p15.33-*TERT* and 6q21-*CD164* were associated with small clone CH. For small clone CH risk, we also identified a previously unreported locus marked by rs72755524 at 5p13.3 in a region with several long non-coding RNAs (lncRNAs) (Fig. 4e and Supplementary Table 21). Additional signals suggested by approximate conditional analysis at each locus identified in this study are listed in Supplementary Table 17. Examining heterogeneity of associations across the five CH traits using forest plots (Extended Data Fig. 5a–e and Supplementary Tables 16 and 18–21) revealed that in addition to 14q32.13-*TCL1A*, the lead alleles at 6q21-*CD164* also had opposite effects on *DNMT3A*- versus *TET2*-CH. The lead variants at 6q21-*CD164* and 5p13.3-*LINC02064* were associated with small, but not large, clones while the association at 7q32.2-*TMEM209* was highly specific to *TET2*-CH. The lead variants at 1q42.12-*PARP1* and 3q25.33-*SMC4* had greater effects on large than small clone CH. At the whole-genome level, we estimated the genetic correlation ($r_g$) between *DNMT3A*-CH and *TET2*-CH

as −0.48 (s.e. = 0.33, $P = 0.15$) and large and small clone CH as 0.37 (s.e. = 0.18, $P = 0.018$) using high-definition likelihood inference[30].

Finally, we also performed a focused scan to explore rare variant (MAF: 0.2–1%) associations with the three CH traits with largest case numbers (overall, DNMT3A and small clone CH; each compared with 173,918 controls). This identified one new locus at 22q12.1-CHEK2 where the T allele (frequency = 0.3%) of lead variant rs62237617 was perfectly correlated ($r^2 = 1$) with the 1100delC CHEK2 protein-truncating allele (rs555607708) and conferred a large increase in risk of DNMT3A mutation-associated CH (OR = 4.1, 95% confidence interval (95% CI): 2.7–6.1, $P = 6.3 \times 10^{-12}$).

**Replication of genome-wide significant associations.** Replication was undertaken using independent somatic mutation calling and germline association analysis pipelines on data from 221,285 European-ancestry individuals in the UKB, for whom WES was performed after our UKB discovery set. We focused on DNMT3A and/or TET2 mutation carriers ($n = 9,386$) in the replication sample, stratified by these two genes and clone size, and evaluated the 20 unique lead variants identified in the discovery GWAS (representing 26 distinct overall/subtype-specific CH associations). Eighteen of 20 variants were replicated at $P < 0.05$, with 16 replicating at $P < 0.0025$ (accounting for testing 20 variants), and 19 showing consistent directionality (Supplementary Table 22). Variants rs13130545 (overall CH; 4q35.1-ENPP6) and rs72755524 (small clone CH; 5p13.3-LINC02064) were not associated at $P < 0.05$ in replication analysis. Notably, we confirmed our observation that lead alleles at TCL1A and CD164 had opposite effects on DNMT3A- and TET2-CH, and replicated the CHEK2 association.

**Blood chromosomal mosaicism and CH due to gene mutation.** It is not known whether the germline genetic architecture underlying predisposition to CH due to individual gene mutations is similar to that underlying CH due to mosaic chromosomal alterations (mCAs). We used data from a recent blood mCA GWAS[31] to answer this and found that 13 of 19 unique lead variants identified for the five gene-mutant CH traits were associated with hematological mCA risk ($P < 10^{-4}$; Supplementary Table 23). Notably, for our lead variants rs2296312 (14q32.13-TCL1A) and rs8088824 (18q12.3-SETBP1), the alleles conferring increased DNMT3A-CH risk reduced hematological mCA risk (Supplementary Table 23). We found a correlation between overall CH and mCAs ($r_g = 0.44$, s.e. = 0.21, $P = 0.037$) using LDSC[23]. This germline genetic correlation together with enrichment of the CH GWAS signal in common lymphoid and myeloid progenitors (Fig. 3b) supports the recent finding that gene-mutant CH and mCAs have overlapping biology that leads them to confer risk of both lymphoid and myeloid malignancies[32]. Further, a phenome-wide scan[33,34] showed that several newly identified lead variants in our analyses were associated with multiple blood cell counts/traits (Supplementary Table 24).

**Gene-level associations and network analyses.** We used two complementary methods to perform gene-level association tests for each of our five CH traits: multi-marker analysis of genomic annotation (MAGMA) and a transcriptome-wide association study using blood-based *cis* gene expression quantitative trait

locus (eQTL) data on 31,684 individuals[35] and summary-based Mendelian randomization (SMR) coupled with the heterogeneity in dependent instruments (HEIDI) colocalization test[36]. Both approaches converged on a new locus at 6p21.1, associated at gene-level genome-wide significance ($P_{MAGMA} < 2.6 \times 10^{-6}$, $P_{SMR} < 3.2 \times 10^{-6}$) with DNMT3A-CH and marked by CRIP3 ($P_{MAGMA} = 3.4 \times 10^{-7}$, $P_{SMR} = 6.6 \times 10^{-7}$; Fig. 5a and Supplementary Tables 25 and 26). While CRIP3 was the only 6p21.1 gene to reach gene-level genome-wide significance in both MAGMA and SMR, we did find subthreshold evidence for association between SRF or ZNF318 in the same region and DNMT3A-CH (Fig. 5a). Notably, SRF encodes the serum response factor known to regulate HSC adhesion[37] while ZNF318 is an occasional CH somatic driver[38]. More globally, protein–protein interaction (PPI) network analysis[39], using proteins encoded by the 57 genes with $P_{MAGMA} < 0.001$ in the overall CH analysis (Supplementary Table 25) as 'seeds', identified the largest subnetwork (Fig. 5b) as encompassing 13 of 57 proteins with major hub nodes highlighted as TERT, PARP1, ATM and SMC4. This was consistent with the emerging theme that potential trait-associated genes at subthreshold GWAS loci are often part of interconnected biological networks[40,41]. The subthreshold genes identified by MAGMA that encoded protein hubs in this network included FANCF (DNA repair pathway) and PTCH1 (hedgehog signaling; Fig. 5b), both implicated in acute myeloid leukemia pathogenesis[42,43], and GNAS, a CH somatic driver[44]. The CH subnetwork was significantly enriched for several pathways including DNA repair, cell cycle regulation, telomere maintenance and platelet homeostasis (Supplementary Table 27).

**Functional target gene prioritization at CH risk loci.** To prioritize putative functional target genes at $P_{lead-variant} < 5 \times 10^{-8}$ loci identified by our GWAS of five CH traits, we combined gene-level genome-wide significant results from MAGMA and SMR (Supplementary Tables 25 and 26) with five other lines of evidence: PPI network hub status (Supplementary Table 28); variant-to-gene searches of Open Targets[45] for lead variants; and overlap between fine-mapped variants[46,47] (Supplementary Table 29) and (1) gene bodies, (2) regions with accessible chromatin correlated with nearby gene expression in hematopoietic progenitor cells[25,48–50] and (3) missense variant annotations[51,52] (Supplementary Table 30). The genes nominated by the largest number of approaches, representing the most likely targets, were SMC4, ENPP6, TERT, CD164, ATM, PARP1, TCL1A, SETBP1 and TMEM209 (Supplementary Table 31).

Among the newly identified loci, lead variant rs138994074 at 1q42.12 was strongly correlated ($r^2 = 0.93$) with rs1136410, a missense germline mutation in PARP1 (Supplementary Table 30) wherein the G allele, which is protective for DNMT3A-CH, leads to a missense variant (p.Val762Ala) in the catalytic domain of its protein product associated with reduced Poly(ADP-ribose) polymerase 1 activity[53]. While SETBP1 was the only gene nominated at 18q12.3 (by only one approach, Open Targets[45]), its nomination is strengthened by the fact that somatic SETBP1 mutations are recognized drivers of myeloid malignancies[54,55]. We also evaluated the 'druggability' of the prioritized genes in the context of known therapeutics (yielding support for TERT and PARP1) and ongoing drug development (yielding limited support for SMC4, ATM,

**Fig. 4 | Manhattan plots displaying genome-wide associations between common germline genetic variants and each of five CH traits.** The y axes depict P values ($-\log_{10}$) for associations derived from the noninfinitesimal mixed model association test implemented in BOLT-LMM. The x axes depict chromosomal position on build 37 of the human genome (GRCh37). The dotted lines indicate the genome-wide significance threshold of $P = 5 \times 10^{-8}$. Known (previously published) and new loci are indicated by cytoband and target gene (based on the prioritization exercise described in the text). Since there were multiple independent loci at 5p15.33 (LD $r^2 < 0.05$), we also label the 5p15.33 signals using the lead variant rs number for each signal. Our prioritization exercise was focused on protein coding genes near each lead variant and since there were no protein coding genes within 1 Mb of the lead variant at 5p13.3, we labeled this association using the nearest noncoding RNA. The CH traits corresponding to each Manhattan plot are: **a**, Overall CH. **b**, CH with mutant DNTM3A. **c**, CH with mutant TET2. **d**, CH with large clones. **e**, CH with small clones.

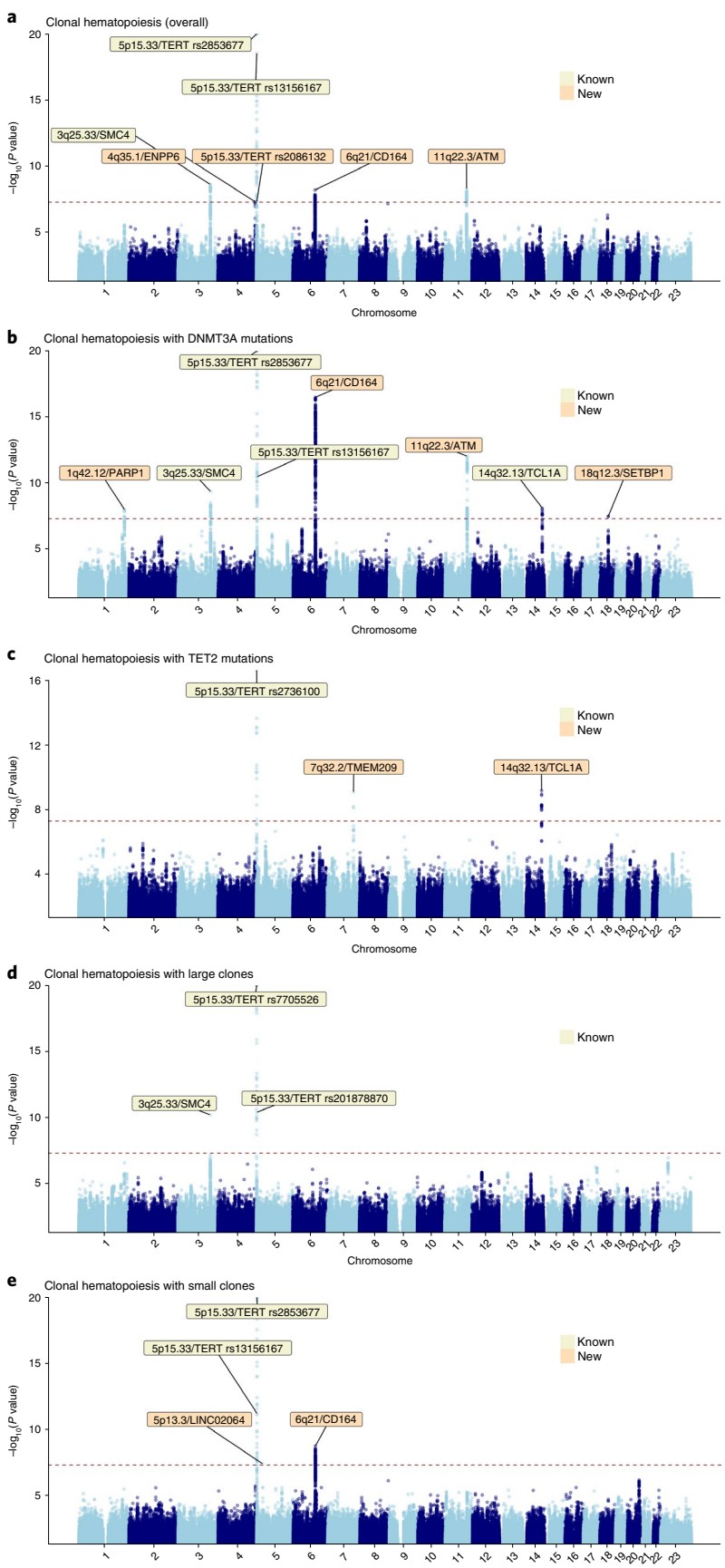

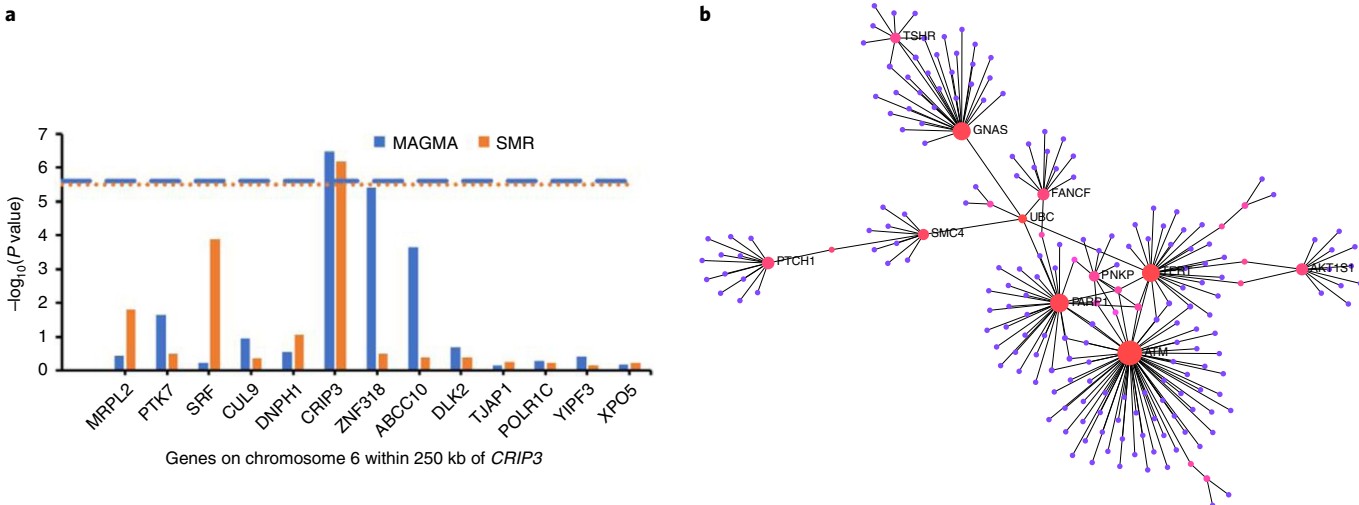

**Fig. 5 | Gene-level association and PPI network analyses. a**, Gene-level associations in the 6q21 region within 25 kb of *CRIP3*, that is, between GRCh37 positions 43,017,448 and 43,526,535 on chromosome 6. The *x* axis lists all the genes in this region that were tested by both MAGMA and SMR. MAGMA uses a multiple linear principal components regression model while SMR is based on the Wald test. *CRIP3* was the only gene located more than 1 Mb away from a GWAS-identified lead variant that was found to be associated with CH at gene-level genome-wide significance by both MAGMA and SMR. The *y* axis depicts the *P* value (−log₁₀) for association in the MAGMA and SMR analyses. The gene-level genome-wide significance threshold in MAGMA ($P = 2.6 \times 10^{-6}$ after accounting for 19,064 genes tested) is indicated by the blue dashed line and in SMR ($P = 3.2 \times 10^{-6}$ after accounting for 15,672 genes tested) by the orange dotted line. Both *CRIP3* and *SRF* had SMR HEIDI $P > 0.05$ indicating colocalization of the GWAS and eQTL associations. The HEIDI test is a test of heterogeneity of Wald ratio estimates. **b**, Largest subnetwork of genes/proteins associated with overall CH risk identified by the NetworkAnalyst tool. NetworkAnalyst uses a 'Walktrap' random walks search algorithm to identify the largest first-order interaction network. All genes (*n* = 57) with $P_{\text{MAGMA}} < 0.001$ in the overall CH MAGMA analysis were mapped to proteins and used as 'seeds' for network construction which was done by integrating high-confidence PPIs from the STRING database. The largest subnetwork constructed contained 13 of the 57 seed proteins and included 210 nodes and 231 edges. The colored nodes indicate seed proteins that interact with at least two other proteins in this subnetwork with the intensity of redness increasing with number of interacting proteins. Seed proteins that interact with six or more other proteins in the subnetwork are named above their corresponding node.

*CD164* and *ENPP6*) using the Open Targets[56] and canSAR[57] databases (Supplementary Tables 32 and 33).

**Mendelian randomization (MR) analyses.** We integrated several large GWAS datasets (Supplementary Tables 34 and 35) and used inverse-variance-weighted (IVW) MR[58] to appraise putative causes and consequences of CH. Genetically predicted smoking initiation[59] was associated with overall CH risk (OR = 1.15, 95% CI: 1.05–1.25, $P = 2.2 \times 10^{-3}$). Point estimates of effect sizes were consistent in direction across MR analyses for *DNMT3A*, *TET2*, and large and small clone CH (Fig. 6a and Supplementary Table 36), with the largest OR observed for large clone CH (OR = 1.24). We also appraised the roles of leukocyte telomere length (LTL)[60], alcohol use[59], adiposity[61], genetic liability to T2D (ref. [62]), circulating lipids[63], blood-based epigenetic aging phenotypes[64], blood cell counts/indices[29] and circulating cytokines/growth factors[65] as potential risk factors for CH (Fig. 6 and Supplementary Tables 36–38 for full results, including sensitivity analyses). Genetically predicted longer LTL was associated with increased overall CH risk (OR = 1.56, 95% CI: 1.25–1.93, $P = 5.7 \times 10^{-5}$), and with *DNMT3A*-, *TET2*-, and large and small clone CH (Fig. 6b and Supplementary Table 36). Higher genetically predicted body mass index (BMI) was associated with increased risk of large clone CH (OR = 1.15, 95% CI: 1.01–1.31, $P = 0.029$). Genetically elevated circulating apolipoprotein B levels were associated with increased (OR = 1.18, 95% CI: 1.01–1.36, $P = 0.032$; Fig. 6c), whilst genetically predicted alcohol use was associated with decreased (OR = 0.46, 95% CI: 0.25–0.83, $P = 0.010$), risk of *TET2*-CH. Among cytokines, genetically elevated circulating macrophage inflammatory protein 1a, a regulator of myeloid differentiation and HSC numbers[66], was associated with risk of *DNMT3A*-CH (OR = 1.13, 95% CI 1.03–1.23, $P = 7.1 \times 10^{-3}$; Supplementary Table 38).

We used independent ($r^2 < 0.001$) variants associated with overall, *DNMT3A*, *TET2*, and large and small clone CH at $P < 10^{-5}$ as genetic instruments for each of these traits and assessed their associations with outcomes (Supplementary Tables 35, 39 and 40). Since more variants were available at $P < 5 \times 10^{-8}$ for overall and *DNMT3A*-CH, we also examined the consistency of associations when using genome-wide (GWS; $P < 5 \times 10^{-8}$) and sub-genome-wide significant (sub-GWS; $P < 10^{-5}$) instruments for these two traits. Using the sub-GWS instrument, genetic liability to overall CH had the largest associations (Fig. 7a) with myeloproliferative neoplasms (MPN) risk[48] (OR = 1.99, 95% CI: 1.23–3.23, $P = 5.4 \times 10^{-3}$), intrinsic epigenetic age acceleration[64] (which represents a core characteristic of HSCs[67]; beta = 0.39, 95% CI: 0.08–0.69, $P = 0.01$) and the blood-based Hannum epigenetic clock[64] (beta = 0.27, 95% CI: 0.04–0.49, $P = 0.02$) and even larger associations were observed when using the GWS instrument. Genetic liability to CH conferred increased risks of lung[68], prostate[69], ovarian[70], oral cavity/pharyngeal[71] and endometrial cancers[72] (Fig. 7a,b and Supplementary Table 39). MR analyses did not support causal risk-conferring associations between genetic liability to CH and CAD[73], ischemic stroke[74] and heart failure[75], with similar lack of evidence across gene-specific and clone size-specific CH, and GWS instrument analyses (Fig. 7a,b and Supplementary Table 39). However, we did uncover an association between genetic liability to overall CH or *DNMT3A*-CH and atrial fibrillation[76] risk (OR = 1.09, 95% CI: 1.04–1.15, $P = 4.9 \times 10^{-4}$ for overall CH with the GWS instrument; Supplementary Table 39). Among cytokines/growth factors[65], genetic liability to overall CH was associated with elevated circulating stem cell growth factor beta (beta = 0.19; 95% CI: 0.07–0.30, $P = 1.1 \times 10^{-3}$). MR analyses also revealed bidirectional associations between CH phenotypes and several blood cell counts/traits[29], suggesting a shared heritability (Figs. 6b and 7a,b and Supplementary

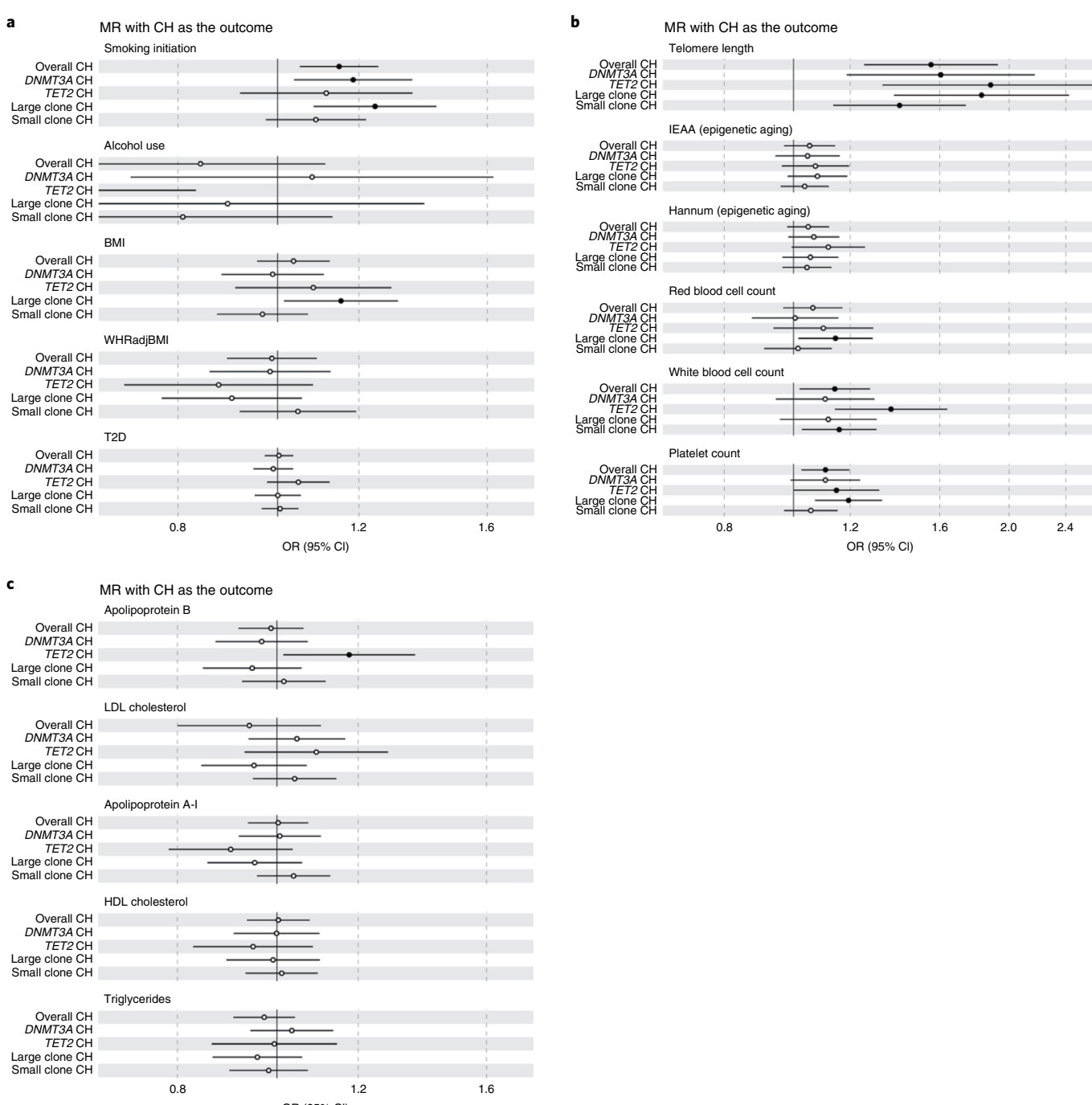

**Fig. 6 | IVW MR forest plots with CH traits as outcomes. a–c**, ORs for CH risk are represented as per (1) standard deviation unit for continuous exposures (alcohol use in drinks per week, BMI, waist-to-hip ratio adjusted for BMI (WHRadjBMI) (**a**); LTL, two epigenetic aging traits, and red cell, white cell and platelet counts (**b**); and five circulating lipid traits (**c**)) and (2) log-odds unit for binary exposures (smoking initiation (ever having smoked regularly) and genetic liability to T2D (**a**)). IVW regression was used for all MR analyses, and results were not adjusted for multiple comparisons. Details of units are provided in Supplementary Table 34. Symbols represent OR markers, and OR marker symbols with corresponding *P* < 0.05 are represented by filled circles. Error bars represent 95% CIs. Sample sizes for the smoking, alcohol, BMI, WHRadjBMI, T2D, apolipoproteins B and A-I, LDL, HDL and triglycerides analyses are provided in Supplementary Table 34. Sample sizes for the LTL, IEAA, Hannum and three blood cell count analyses are provided in Supplementary Table 35. Full results, including from sensitivity analyses, are presented in Supplementary Tables 36–38. WHRadjBMI, waist-to-hip ratio adjusted for BMI; LDL, low-density lipoprotein cholesterol; IEAA, intrinsic epigenetic age acceleration.

Tables 37 and 39). We found little evidence to support an association between genetic liability to CH and LTL (Supplementary Table 39). Finally, we also performed an MR-PheWAS evaluating associations between genetic liability to overall or *DNMT3A*-CH and 1,434 disease/trait outcomes in the UKB. Reassuringly, the strongest associations involved blood cell counts/traits and hematopoietic cancers, but we also uncovered new associations such as with malignant skin cancers (Supplementary Tables 41 and 42). Results of MR sensitivity analyses using the weighted median[77] and MR-Egger[78] methods are provided in Supplementary Tables 36–42.

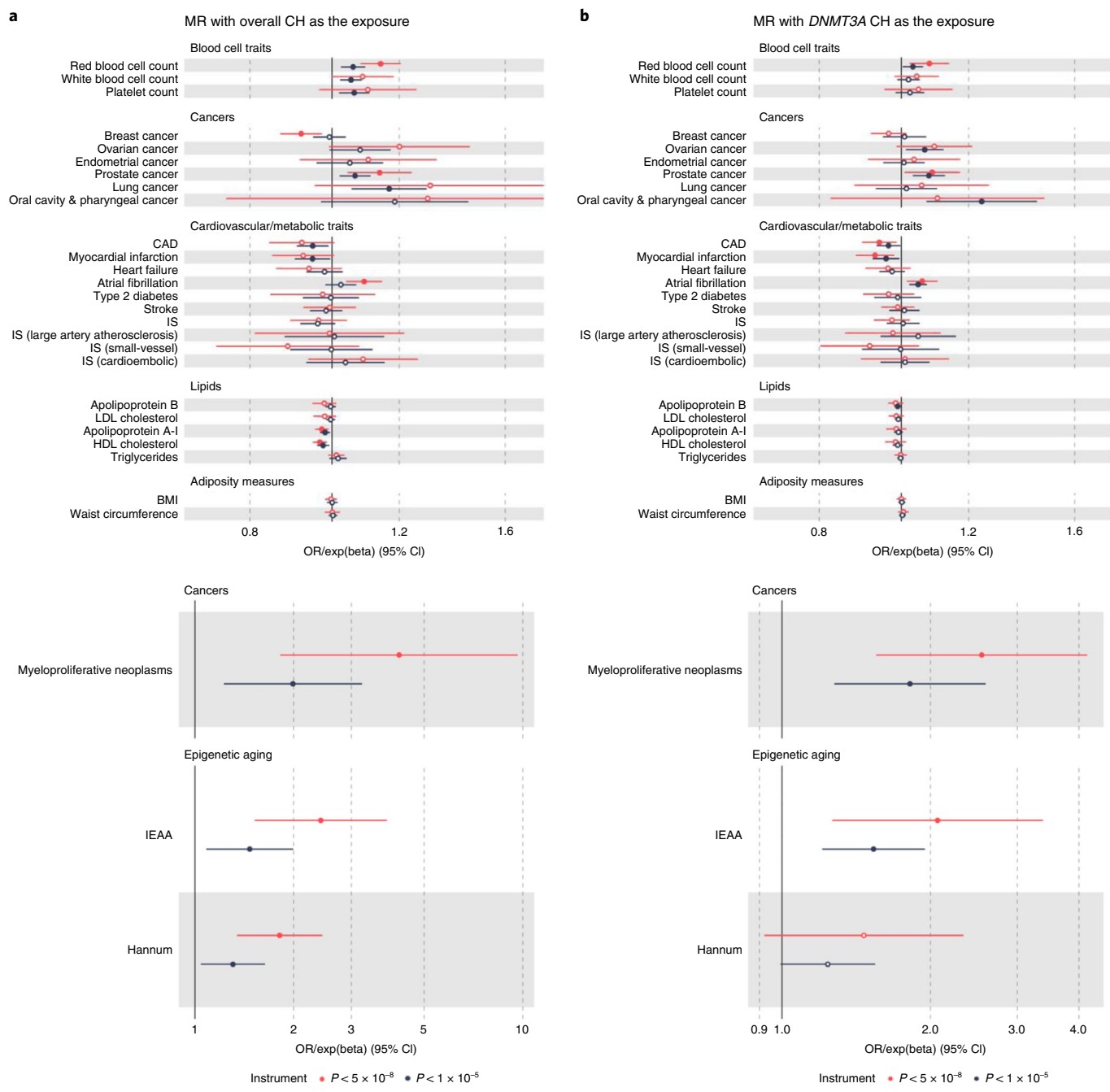

**Fig. 7 | IVW MR forest plots with CH traits as exposures.** Forest plots with OR markers (for cancers and cardiovascular/metabolic traits) or exponentiated beta coefficient (exp(beta)) markers (for blood cell traits, lipids, adiposity measures and epigenetic aging indices). ORs/exp(betas) are represented as per log-odds unit increase in genetic liability to overall CH (**a**) or *DNMT3A*-CH (**b**). OR/exp(beta) markers with corresponding $P < 0.05$ are represented by filled circles. IVW regression was used for all MR analyses, and results were not adjusted for multiple comparisons. Symbols represent OR markers and error bars represent 95% CIs. Red symbols and error bars represent results using genetic instruments comprised exclusively of genome-wide significant ($P < 5 \times 10^{-8}$) variants. Black symbols and error bars represent results when using genome-wide significant and sub-GWS ($P < 10^{-5}$) variants in the genetic instrument. Large effect size estimates (ORs/exp(betas)) are shown in the lower panels. Sample sizes for all genome-wide association datasets used are provided in Supplementary Table 35. Full results, including from sensitivity analyses, are presented in Supplementary Tables 39 and 40. IS, ischemic stroke.

## Discussion

We present an observational and genetic epidemiological analysis of CH in 200,453 individuals in the UKB and report a series of insights into the causes and consequences of this common aging-associated phenomenon. We increase the number of germline associations with CH in European-ancestry populations from 4 (ref. [17]) to 14, reveal heterogeneity of associations by CH driver gene and clone size, and implicate putative new CH susceptibility genes, including

*CD164*, *ATM* and *SETBP1*, through functional annotation. We also demonstrate that the CH GWAS signal is enriched at epigenetic marks specific to the hematopoietic system, particularly in open chromatin regions of hematopoietic stem/progenitor cells. The robustness of our GWAS is supported by replication of the vast majority of associations in an additional set of 221,285 individuals from the UKB and further affirmed by our replication of previous European-ancestry-specific CH associations[17], the consistency of our estimates of CH heritability

with previous reports[17,79] and the fact that many of our lead variants are associated with related traits[29,31,60,80].

At 14q32.13-*TCL1A*, we replicate the reported association with *DNMT3A*-CH (ref. [17]) and identify a new genome-wide significant association with *TET2*-CH. Strikingly, however, we found that the association operates in the opposite direction for *TET2*-CH, versus *DNMT3A*-CH. This inverse relationship, also supported by our finding of a suggestive negative genetic correlation between *TET2*- and *DNMT3A*-CH, is tantalizing in light of recent observations that ageing has different effects on the dynamics of these two forms of CH, resulting in *TET2*-CH becoming more prevalent than *DNMT3A*-CH in those over 80 yr (refs. [20,81]). Also notable in this light is the finding of an association at 6q21-*CD164* with *DNMT3A*-CH, and a trend in the opposite direction for *TET2*-CH that was confirmed in the replication analysis. As *CD164* is expressed in the earliest HSCs[82] and encodes a key regulator of HSC adhesion[83,84], this proposes that HSC migration and homing may play important roles in CH pathogenesis. The reciprocal relationship of both *TCL1A* and *CD164* with the two main CH subtypes suggests that their expression must be tightly regulated to prevent the development of one or other CH subtype, making these loci important targets for hijack by the effects of somatic mutations. In fact, a recent study[85] suggests that this may be how *TET2* and *ASXL1* mutations interact with a *TCL1A* promoter variant associated with clonal expansion rate. *TCL1A* is not expressed in normal or *DNMT3A*-mutated HSCs and the authors show that the locus becomes susceptible to activation in the presence of *TET2* or *ASXL1* mutations only when harboring the reference allele at the promoter variant, leading to faster clonal expansion. This type of interaction may operate for *CD164* and other CH risk loci, or alternative models of interaction between the germline and somatic genome may exist.

New CH risk loci included the *PARP1* coding variant rs1136410, where the G allele is protective for *DNMT3A*-CH and associated with reduced catalytic activity[53] suggesting that this most common form of CH may be vulnerable to PARP inhibition, in keeping with the observed synergy between PARP and DNMT inhibitors[86]. We also identified three lead variants at the *TERT* locus for which CH risk alleles were associated with longer LTL, a finding corroborated by our MR results linking increased LTL to CH. Interestingly, a recent study found deleterious rare germline *TERT* variants associated with shorter telomeres in patients with myelodysplastic syndromes[87]. However, compared with conventional myelodysplastic syndromes, these cases displayed a paucity of somatic mutations in *DNMT3A* (2 of 41) and *TET2* (3 of 41 cases), suggesting that evolutionary paths may differ between cases with long versus short telomeres.

The rich phenotypic data captured by the UKB, coupled with our genetic analysis of CH and external GWAS datasets, enabled us to explore associations of CH using multivariable regression and interrogate, at scale, potential causal relationships between CH and its putative risk factors and consequences using MR. This highlighted that smoking and longer telomere length are causal risk factors for CH. These associations were valid across multiple CH subtypes and, in the case of smoking, corroborated by observational estimates. We also reveal that not only is genetic predisposition to CH causally associated with MPN risk, but it also increases the risk of lung, prostate, ovarian, oral/pharyngeal and endometrial cancers. In these analyses, the use of two-sample MR protected against potential reverse causality arising from cancer therapy-induced selection pressure on hematopoietic clones[88]. These MR results suggest that genetic liability to CH may be a biomarker for development of cancer elsewhere in the body, analogous to the link between genetic predisposition to Y chromosome loss in blood and solid tumor risk[89].

We investigated the recently identified association of CH with blood-based epigenetic clocks[90], using bidirectional MR, and show that this association is likely to be causal in the direction from CH to epigenetic age acceleration. We also showed that genetic predisposition to CH was associated with elevated circulating levels of stem cell growth factor beta, a secreted sulfated glycoprotein that regulates primitive hematopoietic progenitor cells[91]. Finally, we unraveled a previously unreported association between genetic liability to CH and atrial fibrillation risk, which was also supported by our observational analysis. However, unlike previous reports based on smaller sample sizes[5,10,22], we did not find evidence in observational and MR analyses to support an association between CH and CAD or ischemic stroke. However, our MR analyses indicated that higher BMI and circulating apolipoprotein B levels were associated with *TET2* and large clone CH risks, respectively, with apolipoprotein B being the key causal lipid risk factor for CAD[63,92]. We also demonstrated the impact of age, in particular, as a strong confounder of the CH–CAD/ischemic stroke associations. These results raise the possibility that reported associations of CH with CAD/stroke risks may suffer from residual confounding. Moreover, many of the cohorts that reported these associations are enriched in participants at high cardiovascular risk[10], in contrast to the UKB, where participants may be healthier, and potentially have lower epigenetic aging. Recent findings suggest that CH is associated with CAD/stroke only on a background of epigenetic aging[90], offering a plausible mechanistic explanation for the absence of an association in our study.

Collectively, our findings substantially illuminate the landscape of inherited susceptibility to CH and provide insights into the causes and consequences of CH with implications for human health and ageing.

## Online content

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

## Methods

**Study population and WES data.** The UKB resource was approved by the North West Multi-centre Research Ethics Committee under reference number 21/NW/0157 and all participants provided written, informed consent to participate. Participants in the UKB are volunteers and not compensated for participation. Data from the UKB resource were accessed under approved application numbers 56844, 29202 and 26041 for this study. The UKB is a prospective longitudinal study containing in-depth genetic and health information from half a million UK participants. For this study, we have selected 200,453 individuals (200k) who had WES data available (age range: 38–72, median age: 58; 55% females). WES was generated in two batches, the first of approximately 50,000 samples (50k)[93] and the second comprising an additional 150,000 samples (150k)[19]. Exomes were captured using the IDT xGen Exome Research Panel v.1.0 including supplemental probes; a different IDT v.1.0 oligo lot was used for each batch. Multiplexed samples were sequenced with dual-indexed 75×75-base-pair paired-end reads on the Illumina NovaSeq 6000 platform using S2 (50k samples) and S4 (150k samples) flow cells. The 50k samples were first computed using FE protocol and reprocessed later to match the second batch of 150k sequences which were processed using a new improved unified OQFE pipeline. As the initial 50k samples were sequenced on S2 flow cells and with a different IDT v.1.0 oligo lot from the remaining 150k samples, which were sequenced on S4 flow cells, we included the WES batch as a covariate in downstream analyses.

**Sequence data processing, CH mutation calling and filtering.** CRAM files generated by the OQFE pipeline were obtained from UKB (Fields 23143-23144). Variant calling on WES data from 200,453 individuals was performed using Mutect2, Genome Analysis Toolkit (GATK) v.4.1.8.1 (ref. [94]). Briefly, Mutect2 was run in 'tumor-only' mode with default parameters, over the exons of 43 genes previously associated with CH (Supplementary Table 1). To filter out potential germline variants we used a population reference of germline variants generated from the 1000 Genomes Project (1000GP)[95] and the Genome Aggregation Database (gnomAD)[96]. All resources were obtained from the GATK Best practices repository (gs:// gatk-best-practices/somatic-hg38). Raw variants called by Mutect2 were filtered out with *FilterMutectCalls* using the estimated prior probability of a reading orientation artifact generated by *LearnReadOrientationModel* (GATK v.4.1.8.1). Putative variants flagged as 'PASS' using FilterMutectCalls or flagged as 'germline' if present at least two times with the 'PASS' flag in other samples were selected for filtering. Gene annotation was performed using Ensembl Variant Effect Predictor (VEP) (v.102)[97]. We required variants with a minimum number of alternate reads of 2, evidence of the variant on both forward and reverse strands, a minimum depth of 7 reads for single nucleotide variants (SNVs) and 10 reads for short indels and substitutions, and a MAF lower than 0.001 (according to 1000GP phase 3 and gnomAD r2.1). For new variants, not previously described in the Catalogue of Somatic Mutations in Cancer (COSMIC; v.91)[98] nor in the Database of Single Nucleotide Polymorphisms (dbSNP; build 153)[99], we used a minimum allele count per variant of 4, and a MAF lower than $5 \times 10^{-5}$. From resulting variants, we selected those that: (1) are included in a list of recurring hotspot mutations associated with CH and myeloid cancer (Supplementary Table 2); (2) have been reported as somatic mutations in hematological cancers at least seven times in COSMIC; or (3) met the inclusion criteria of a predefined list of putative CH variants[17,79] (Supplementary Table 3). We included previous variants flagged as germline by *FilterMutectCalls* if: (1) the number of cases in the cohort flagged as germline was lower than the ones flagged as PASS; and (2) at least one of the cases had a $P < 0.001$ for a one-sided exact binomial test, where the null hypothesis was that the number of alternative reads supporting the mutation was 50% of the total number of reads (95% for copy number equal to one), except for hotspot mutations which were all included. For the final list, we excluded all variants not present in COSMIC or in the list of hotspots that had a MAF equal to or higher than $5 \times 10^{-5}$ and either the mean VAF of all cases was higher than 0.2 or the maximum VAF was lower than 0.1. Frameshift, nonsense and splice-site mutations not present in COSMIC or in the hotspot list were further excluded if for each variant none of the cases had a $P < 0.001$ for a one-sided exact binomial test. A complete list of filtered variants is provided in Supplementary Table 4.

**Trait selection and modeling for observational analyses.** Phenotypes were downloaded in December of 2020 and individual traits were pulled out from the whole phenotype file. Cancer, metabolic and CVD traits were generated, combining individual traits and diagnosis dates based on disease definitions (Supplementary Table 9). For each definition of disease, the first diagnosis event that occurred in each trait was selected. Baseline was defined as the date of sample collection when the individuals attended the assessment centers. The prevalent cases are those identified before the baseline, while incident cases were defined as the events that occurred after the baseline. Unless specified, all regression models included age, sex, smoking status, WES batch and the first ten ancestry principal components as covariates and all analyses were adjusted for multiple comparisons using the false discovery rate (FDR) computed by the Benjamin–Hochberg procedure implemented in the p.adjust function (R stats package v.4.0.2). Blood cell counts and biochemical traits were log10 transformed and analyzed using a logistic regression model with overall and gene-specific CH as outcomes, including the assessment center as covariate and, in the case of cholesterol and cholesterol species, the use of cholesterol-lowering medication as an additional covariate. Individuals with myeloid malignancies or

hematological neoplasms at baseline (that is, with a cancer diagnosis date before the date they attended the assessment centers) were excluded from the analysis. For cancer, CVD and death risk, we performed a time-to-event regression analysis. In the case of cancer and CVD, we performed a competing risk analysis, using the date of death by other cause as the competing event, while for the risk of death we used the Cox proportional hazards model. The cancer/CVD/death event was used as an outcome and CH was considered as the exposure in these analyses. Individuals without the event who died before the end of the follow-up were censored at the time of death, while the rest were censored at the end of the follow-up. For CVD and death risk analyses, we also included BMI, high-density lipoprotein cholesterol, low-density lipoprotein cholesterol, triglycerides, T2D status and hypertension status as covariates. Individuals with myeloid or other malignant neoplasms at baseline were excluded from all aforementioned analyses. The proportional hazards assumption for the Cox and competing risk models was assessed by examining the Schoenfeld residuals. For the phenome-wide association analysis between International Statistical Classification of Diseases and Related Health Problems 10th Revision (ICD-10) codes as outcomes and CH status, logistic regression models were used including age, sex, WES batch and the first ten genetic ancestry principal components as covariates. Analyses were performed over 11,787 selected ICD-10 codes corresponding to disease conditions (A to N), symptoms, signs, and abnormal clinical and laboratory findings (R), and factors influencing health status (Z). All analyses were performed using glm (R stats package v.4.0.2), coxph (R survival package v.3.2-11) and crr (R cmprsk package v.2.2-10) functions.

**Genome-wide association analyses.** Germline genotype data used were from the UKB release that contained the full set of variants imputed into the Haplotype Reference Consortium[100] and UK10K + 1000GP (ref. [95]) reference panels and genotyped on the UK BiLEVE Axiom Array or UKB Axiom Array[101]. Derivation of the analytic sample for UKB individuals of European ancestries followed the QC protocol of Astle et al.[29] and included the following steps: after filtering genetic variants (call rate ≥ 99%, imputation quality info score > 0.9, Hardy–Weinberg equilibrium $P \geq 10^{-5}$) and participants (removal of genetic sex mismatches), we excluded participants having non-European ancestries (self-report or inferred by genetics) or excess heterozygosity (>3 s.d. from the mean), and included only one of each set of related participants (third-degree relatives or closer). After QC, we were left with 10,203 individuals with CH and 173,918 individuals without CH. The subset with CH included 5,185 and 2,041 individuals with *DNMT3A*- and *TET2*-mutant CH, respectively, and 4,049 and 6,154 individuals with large (VAF ≥ 0.1) and small (VAF < 0.1) clone size CH, respectively. Association analyses were performed for autosomal and X chromosomal variants using noninfinitesimal linear mixed models implemented in BOLT-LMM[102] (v.2.3.6) with age at baseline, sex and first ten genetic principal components included as covariates.

Statistically independent lead variants for each CH phenotype were defined using LD-based clumping with an $r^2$ threshold of 0.05 applied across all genotyped and imputed variants, with $P < 5 \times 10^{-8}$, imputation quality score > 0.6 and MAF > 1%. This was implemented using the FUMA pipeline (v.1.3.6b) (ref. [103]). For the rare variant association scan, we used more stringent cut-offs of $P < 10^{-9}$ and imputation quality score > 0.8 to define lead variants but did not require LD-clumping since only one such association was identified. Approximate conditional analysis conditioning on the common (MAF > 1%) lead variants was performed using the --*cojo-cond* flag in the Genome-wide Complex Trait Analysis (GCTA) v.1.93 tool (refs. [27,104]).

We also evaluated associations of the lead variants for overall CH risk in the 505 individuals with CH and 11,893 controls (retained after the QC steps described above), comprising the ancestrally diverse (non-European) subcohort of the 200k UKB cohort, using logistic regression and adjusting for age, sex, WES batch and 40 genetic ancestry principal components.

**Replication of genome-wide significant associations.** Replication analysis was performed using 221,285 unrelated UKB individuals of European ancestry (age range: 39–73, mean age: 57; 53% females), for whom WES was performed subsequent to the initial 200k, using the same protocol. Alignment to the GRCh38 genome reference with Illumina DRAGEN Bio-IT Platform Germline Pipeline v.3.0.7 and QC were performed as detailed by Wang et al.[105]. Somatic variant calling was performed with GATK's Mutect2 (v.4.2.2.0) using a panel of normals to remove recurrent artifacts, and subsequent filtering was performed with *FilterMutectCalls*, including the filtering of read orientation artifacts using priors generated with *LearnReadOrientationModel*. Putative somatic variants were identified from Mutect2 'PASS' calls in *DNMT3A* and *TET2* based on (1) matching the list of putative somatic mutations identified in the discovery cohort, or (2) any *DNMT3A* or *TET2* protein-truncating variants as predefined by Wang et al.[105]. Sample sizes for *DNMT3A*-, *TET2*- and large and small clone *DNMT3A*- or *TET2*-mutant CH are provided in Supplementary Table 22. Replication association statistics were calculated on the 221,285 replication exomes using the imputed genotype data with logistic regression, adopting age, sex and the first four genetic ancestry principal components as covariates.

**Heritability, cell-type enrichment and genetic correlation.** We used LDSC (v.1.0.1)[23] to estimate the narrow-sense heritability of CH on the liability scale

assuming the population prevalence of CH to be 10% (based on the prevalence of CH in the UKB '200k' cohort as shown in Fig. 1b) and constraining the LDSC intercept to 1. The intercept, which in its unconstrained form protects from bias due to population stratification, was constrained to 1 to provide more precise estimates given that there was little evidence of inflation in test statistics due to population structure in unconstrained analysis (unconstrained intercept estimated as 1.009 (s.e. = 0.0067) and lambda genomic control factor of 0.999). We used the pre-computed 1000 Genomes phase 3 European ancestry reference panel LD score dataset for heritability estimation. We used the same LD scores and the --*rg* flag in LDSC to estimate the genetic correlation between the CH and mCA GWAS summary statistics[31]. Cell-type group partitioned heritability analysis was performed using LD scores partitioned across 220 cell-type-specific annotations that were divided into 10 groups[24]: central nervous system, cardiovascular, kidney, adrenal/pancreas, gastrointestinal, connective/bone, immune/hematopoietic, skeletal muscle, liver and other. Each of the ten groups contained cell-type-specific annotations for four histone marks: H3K9ac, H3K27ac, H3K4me1 and H3K4me3 (ref. [24]). We also used LD scores annotated based on open chromatin state (assay for transposase-accessible chromatin using sequencing (ATAC-seq)) profiling by Corces et al.[25,26] in various hematopoietic progenitor cells and lineages at different stages of differentiation. To estimate the genetic correlation between *DNMT3A*- and *TET2*-CH and between large and small clone CH we used the high-definition likelihood (HDL; v.1.4.0)[30] inference approach to improve power given the low sample size in each subtype-specific CH GWAS.

**Gene-level association and network analyses.** We undertook genome-wide gene-level association analyses using two complementary approaches. First, we used MAGMA (v.1.08 implemented in FUMA v.1.3.6b) which involves mapping germline variants to the genes they overlap, accounting for LD between variants and performing a statistical multi-marker association test[106]. Second, we performed a transcriptome-wide association study using blood-based *cis* gene eQTL data on 31,684 individuals from the eQTLGen consortium[35] and SMR coupled with the HEIDI colocalization test to identify germline genetic associations with CH risk mediated via the transcriptome[36]. The gene-level genome-wide significance threshold in the MAGMA analyses was set at $P = 2.6 \times 10^{-6}$ to account for testing 19,064 genes and for SMR was set at $P = 3.2 \times 10^{-6}$ after adjustment for testing 15,672 genes. Further, only genes with SMR $P < 3.2 \times 10^{-6}$ and HEIDI $P > 0.05$ were declared genome-wide significant in the SMR analyses since the HEIDI $P > 0.05$ strongly suggests colocalization of the GWAS and eQTL signals for a given gene[36]. NetworkAnalyst 3.0 (ref. [39]) was used for network analysis. All genes with $P < 10^{-3}$ in each MAGMA analysis for overall, *DNMT3A*- and *TET2*-mutant, and large and small clone CH were used as input. The protein–protein interactome selected was STRING v.10 (ref. [107]) with the recommended parameters (confidence score cut-off of 900 and requirement for experimental evidence to support the PPI). The largest possible network was constructed from the seed genes/proteins and the interactome proteins[39]. Hub nodes were defined as nodes with degree centrality ≥ 10 (that is, a node with at least 10 edges or connections to other proteins in the network as a measure of its importance in the network and consequently its biology). Pathway analysis of this largest network was conducted using the enrichment tool built into NetworkAnalyst and with the Reactome pathway repository therein[108].

**Fine-mapping and target gene prioritization.** We fine-mapped the lead variant signals identified by the FUMA LD-clumping pipeline using the Probabilistic Identification of Causal Single Nucleotide Polymorphisms (PICS2; v.2.1.1) algorithm[46,47] to identify candidate causal variants most likely to underpin each association. The PICS2 algorithm computes the likelihood that each variant in LD with the lead variant is the true causal variant in the region by leveraging the fact that for variants associated merely due to LD, the strength of association scales asymptotically with correlation to the true causal variant[46]. We only retained variants with a PICS2 probability of 1% or more in our final list of fine-mapped candidate causal variants. We overlapped these fine-mapped variants with gene body annotations[48] using GENCODE release 33 (ref. [109]) (build 37) annotations after removing ribosomal protein genes. Fine-mapped variants were also overlapped with ATAC-seq peaks across 16 hematopoietic progenitor cell populations and ATAC-RNA count correlations calculated using Pearson coefficients for hematopoietic progenitor cell RNA counts of genes within 1 Mb of the ATAC peaks and these were used to identify putative target genes of fine-mapped variants that overlapped ATAC-seq peaks[25,48–50]. We also looked up the SIFT[51] and PolyPhen[52] scores for these fine-mapped variants using the SNPnexus v.4 annotation tool[110] to identify coding variants with predicted functional consequences. Finally, we used the Open Targets Genetics resource[45] to identify the most likely target gene of the lead variant at each locus as per Open Targets and used this in our omnibus target gene prioritization scheme described below.

To prioritize putative target genes at the $P_{\text{lead-variant}} < 5 \times 10^{-8}$ loci identified by our GWAS of overall CH, *DNTM3A*-CH, *TET2*-CH and large/small clone size CH, we combined gene-level genome-wide significant results from (1) MAGMA and (2) SMR with (3) PPI network hub status of the gene, (4) variant-to-gene searches of the Open Targets database for lead variants, and overlap between fine-mapped variants and (5) gene bodies, (6) regions with accessible chromatin (ATAC-seq peaks) across 16 hematopoietic progenitor cell populations that were also correlated

with nearby gene expression (RNA sequencing) in the same cell populations and (7) missense variant annotations from SIFT and PolyPhen. Genes nominated by at least two of the seven approaches were listed (except where only one of the seven methods nominated a single gene in a region in which case that gene was listed) and the genes nominated by the largest number of approaches represented the most likely targets at each locus. We also evaluated the 'druggability' of the prioritized functional target genes in the context of known therapeutics and ongoing drug development using the Open Targets Platform[56] and canSAR[57] v.1.5.0 databases. The database canSAR provides chemistry-based (assesses the likely 'ligandability' of a protein based on the chemical properties of compounds tested against the protein itself and/or its homologs) and antibody-based (assesses if a target is potentially suitable for antibody therapy) predictions.

**Phenome-wide association scan for lead variants.** We used PhenoScanner V2 (refs. [33,34]) with catalog set to 'diseases & traits', *P* value set to '5E-8', proxies set to 'EUR' and $r^2$ set to '0.8' to search for published phenome-wide associations between our lead variants or variants in strong LD ($r^2 > 0.8$) with the lead variants and other diseases and traits.

**MR analyses.** MR[111,112] uses germline variants as instrumental variables to proxy an exposure or potential risk factor and evaluate evidence for a causal effect of the exposure or potential risk factor on an outcome. Due to the random segregation and independent assortment of alleles at meiosis, MR estimates are less susceptible to bias from confounding factors as compared with conventional observational epidemiological studies. As the germline genome cannot be influenced by the environment after conception or by preclinical disease, MR estimates are also less susceptible to bias due to reverse causation. MR estimates represent the association between genetically predicted levels of exposures or risk factors and outcomes, as compared with conventional observational epidemiological estimates, which represent direct associations of the exposure or risk factor levels with outcomes. Effect allele harmonization across GWAS summary statistics datasets followed by MR analyses were performed using the TwoSampleMR v.0.5.6 R package[58]. The CH phenotypes were considered as both exposures (to identify consequences of genetic liability to CH) and outcomes (to identify risk factors for CH). When considering CH phenotypes as outcomes, germline variants associated with putative risk factors or exposures at $P < 5 \times 10^{-8}$ were used as genetic instruments for the risk factors/exposures, except for the appraisal of circulating cytokines and growth factors[65] wherein variants associated with cytokines/growth factors at $P < 10^{-5}$ were used as instruments. IVW analysis[113] was the primary analytic approach with pleiotropy-robust sensitivity analyses carried out using the MR-Egger[78] and weighted median[77] methods. A full list of external GWAS data sources used for MR analyses is provided in Supplementary Tables 30 and 31. We also conducted an MR-PheWAS evaluating overall CH and *DNMT3A*-CH as exposures (using variants associated with these at $P < 10^{-5}$) and 1,434 disease and trait outcomes in the UKB data using summary genetic association statistics for the outcomes that were generated by the Neale lab (http://www.nealelab.is/uk-biobank) and accessed via the TwoSampleMR v.0.5.6 R package and the Integrative Epidemiology Unit (IEU) OpenGWAS project portal[114]. FDR control was applied to the MR-PheWAS IVW analysis *P* values.

**Statistics and reproducibility.** No statistical method was used to predetermine sample size. The experiments were not randomized and investigators were not blinded during the experiments and outcome assessment. Participants were excluded from the GWAS due to genetic sex mismatch, excess heterozygosity (>3 s.d. from the mean) and relatedness (only one of each set of participants who were third-degree relatives or closer were retained). To summarize, our study design included observational genomic analyses of CH in 200,453 individuals across ancestries, genome-wide association and post-GWAS analyses for five CH traits (overall, *DNMT3A*, *TET2*, large clone and small clone CH) in 184,121 individuals of European ancestry, followed by trans-ancestry genetic association analyses in 12,398 individuals, and replication genetic association analyses in an additional 221,285 individuals of European ancestry—all from the UKB.

**Reporting summary.** Further information on research design is available in the Nature Research Reporting Summary linked to this article.

## Data availability
Summary statistics for the overall and subtype-specific CH genome-wide association analyses reported here have been made publicly available at https://doi.org/10.5281/zenodo.5893861. They can also be downloaded from the GWAS Catalog (https://ftp.ebi.ac.uk/pub/databases/gwas/summary_statistics/GCST90102001-GCST90103000/) using the study accession numbers GCST90102618 (overall CH; https://www.ebi.ac.uk/gwas/studies/GCST90102618), GCST90102619 (*DNMT3A*-CH; https://www.ebi.ac.uk/gwas/studies/GCST90102619), GCST90102620 (*TET2*-CH; https://www.ebi.ac.uk/gwas/studies/GCST90102620), GCST90102621 (small clone CH; https://www.ebi.ac.uk/gwas/studies/GCST90102621) and GCST90102622 (large clone CH; https://www.ebi.ac.uk/gwas/studies/GCST90102622). Individual-level UK Biobank data can be requested via application to the UK Biobank (https://www.ukbiobank.ac.uk). The CH call set has been returned to the UK Biobank to enable individual-level data

linkage for approved UK Biobank applications. Pre-computed 1000 Genomes phase 3 European ancestry reference panel and cell-type group LD scores used for heritability estimation and cell-type group partitioned heritability analysis, respectively, can be downloaded from https://alkesgroup.broadinstitute.org/LDSCORE. The STRING protein–protein interaction and Reactome pathway databases are available from, and were accessed via, the NetworkAnalyst 3.0 platform (https://www.networkanalyst.ca) and are also downloadable separately from http://version10.string-db.org/cgi/download.pl for STRING v.10 and https://reactome.org/download-data for Reactome. *cis*-expression quantitative trait locus data from the eQTLGen consortium used for the SMR analyses can be downloaded from https://www.eqtlgen.org/cis-eqtls.html. All summary genetic association statistics datasets used in the Mendelian randomization (MR) and MR-phenome-wide association analyses are publicly available at the Integrative Epidemiology Unit (IEU) OpenGWAS project portal (https://gwas.mrcieu.ac.uk/datasets) and can be accessed by entering the identifiers provided in the 'id/link' column in Supplementary Table 35 and the 'id.outcome' column in Supplementary Tables 41 and 42 in the 'GWAS ID' field at https://gwas.mrcieu.ac.uk/datasets.

## Code availability

We used publicly available software and web-based tools for analyses: somatic variant calling was performed using Mutect2 GATK v.4.1.8.1 for the discovery whole-exome sequences and using Mutect2 GATK v.4.2.2.0 for the replication whole-exome sequences (https://github.com/broadinstitute/gatk/releases). Genome-wide association analyses were conducted using BOLT-LMM v.2.3.6 (https://alkesgroup.broadinstitute.org/BOLT-LMM/BOLT-LMM_manual.html). LDSC v.1.0.1 (https://github.com/bulik/ldsc) was used for heritability estimation and cell-type-specific enrichment analyses. Genetic correlations were computed using HDL v.1.4.0 (https://github.com/zhenin/HDL). LD-clumping was undertaken using FUMA v.1.3.6b (https://fuma.ctglab.nl). Conditional analysis was carried out using the GCTA-COJO tool (https://yanglab.westlake.edu.cn/software/gcta/#COJO) in GCTA v.1.93 (https://yanglab.westlake.edu.cn/software/gcta/#Download). Gene-level association analyses were performed using the MAGMA v.1.08 tool (https://ctg.cncr.nl/software/magma) in FUMA v.1.3.6b (https://fuma.ctglab.nl). Transcriptome-wide association analyses were conducted using SMR v.1.03 (https://yanglab.westlake.edu.cn/software/smr/#Download). Protein–protein interaction network and pathway analyses were performed using NetworkAnalyst v.3.0 (https://www.networkanalyst.ca). Fine-mapping was conducted using PICS2 v.2.1.1 (https://pics2.ucsf.edu). Fine-mapped variant-gene body overlap and fine-mapped variant ATAC-seq peak overlap and ATAC-RNA count correlations were performed using the scripts at https://github.com/sankaranlab/mpn-gwas. SIFT and PolyPhen scores were obtained via the SNPnexus v.4 (https://snp-nexus.org). Lead variants were searched on the Open Targets Genetics (https://genetics.opentargets.org) and PhenoScanner V2 (http://www.phenoscanner.medschl.cam.ac.uk) platforms. Druggability of prioritized functional target gene products was evaluated using the Open Targets platform (https://platform.opentargets.org) and canSAR v.1.5.0 (https://cansarblack.icr.ac.uk). For the observational analyses, we used the glm and p.adjust (in the R stats package v.4.0.2), coxph (in the R survival package v.3.2-11) and crr (in the R cmprsk package v.2.2-10) functions, all implemented in R v.4.0.2. For the Mendelian randomization analyses, we used the TwoSampleMR v.0.5.6R package implemented in R v.4.0.5. Custom code was written for variant calling (and made available as a Nextflow v.20.07.1 pipeline) and Mendelian randomization and these scripts are available at https://doi.org/10.5281/zenodo.6419042 (https://github.com/siddhartha-kar/clonal-hematopoiesis).

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

## Acknowledgements

This work was funded by a joint grant from the Leukemia and Lymphoma Society (grant no. RTF6006-19) and the Rising Tide Foundation for Clinical Cancer Research (grant no. CCR-18-500), and by the Wellcome Trust (grant no. WT098051) to G.S.V. This work was also supported by the National Institute for Health Research Cambridge Biomedical Research Centre (grant no. BRC-1215-20014). The views expressed are those of the authors and not necessarily those of the National Institute for Health Research or the Department of Health and Social Care. S.P.K. is supported by a United Kingdom Research and Innovation (UKRI) Future Leaders Fellowship (grant no. MR/T043202/1). P.M.Q. is funded by the Miguel Servet Program (grant no. CP20/00130). M.A.F. is funded by a Wellcome Clinical Research Fellowship (grant no. WT098051). R.L. is supported by Cancer Research UK (grant no. C18281/A29019). P.C. is supported by a British Heart Foundation Clinical Training Research Fellowship. S.B. is supported by a Sir Henry Dale Fellowship jointly funded by the Wellcome Trust and the Royal Society (grant no. 204623/Z/16/Z). G.S.V. is supported by a Cancer Research UK Senior Cancer Fellowship (grant no. C22324/A23015) and work in his laboratory is also funded by the European Research Council, Kay Kendall Leukaemia Fund, Blood Cancer UK and the Wellcome Trust. This research was conducted using the UK Biobank resource under approved applications 56844, 29202 and 26041. We thank the participants and investigators involved in the UK Biobank resource and in the other genome-wide association studies cited in this work who collectively made this research possible.

## Author contributions

S.P.K., P.M.Q. and G.S.V. conceived, designed and supervised the study. S.P.K. and P.M.Q. carried out data analyses and generated tables and figures. M.G., M.S.V. and M.A.F. helped with mutation calling and filtering. T.J. and S.B. performed genome-wide association analyses. J.M. conducted and interpreted the replication analyses. R.L. assisted with Mendelian randomization analyses. V.I. helped with UK Biobank data access and handling. S.B. and P.C. advised on Mendelian randomization analyses. C.B. and P.C. helped with UK Biobank trait selection and filtering. S.P. supervised and interpreted the replication analyses. S.P.K., P.M.Q. and G.S.V. drafted the manuscript with inputs from all authors. All authors approved the final version of the paper.

## Competing interests

G.S.V. is a consultant to STRM.BIO and holds a research grant from AstraZeneca for research unrelated to that presented here. J.M. and S.P. are current employees and/or stockholders of AstraZeneca. The remaining authors declare no competing interests.

## Additional information

**Extended data** is available for this paper at https://doi.org/10.1038/s41588-022-01121-z.

**Correspondence and requests for materials** should be addressed to Siddhartha P. Kar, Pedro M. Quiros or George S. Vassiliou.

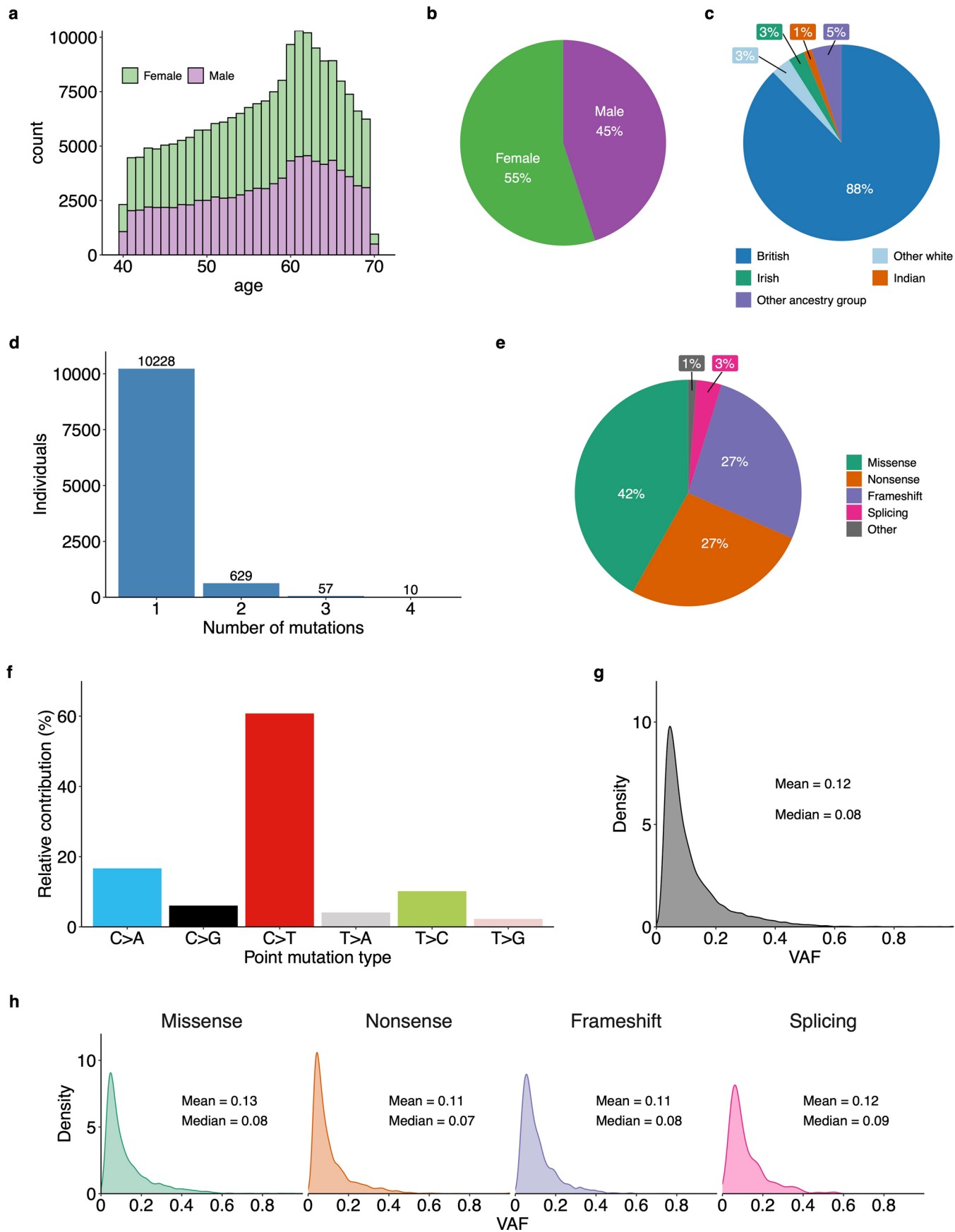

**Extended Data Fig. 1 |** See next page for caption.

**Extended Data Fig. 1 | Characterization of CH in the UK Biobank. a**, Histogram stratified by sex showing the age distribution of individuals in the UKB cohort (n=200,453). **b**, Overall percentage of females and males in the UKB cohort. **c**, Percentage of the most common self-reported ancestry groups in the UKB cohort. Ancestry groups with a frequency lower than 1% were grouped under the 'Other ancestry group' category. **d**, Number of individuals with 1, 2, 3, and 4 somatic mutations. More than 90% of individuals with CH had only one driver mutation identified. **e**, Percentages of different CH mutation types identified. **f**, Relative prevalence of each of the six base substitution types amongst the identified CH mutations. **g**, Density plot showing the variant allele fraction (VAF) distribution of all CH somatic mutations. **h**, Density plot showing similar VAF distribution for different mutation types. Mean and median are indicated for g and h.

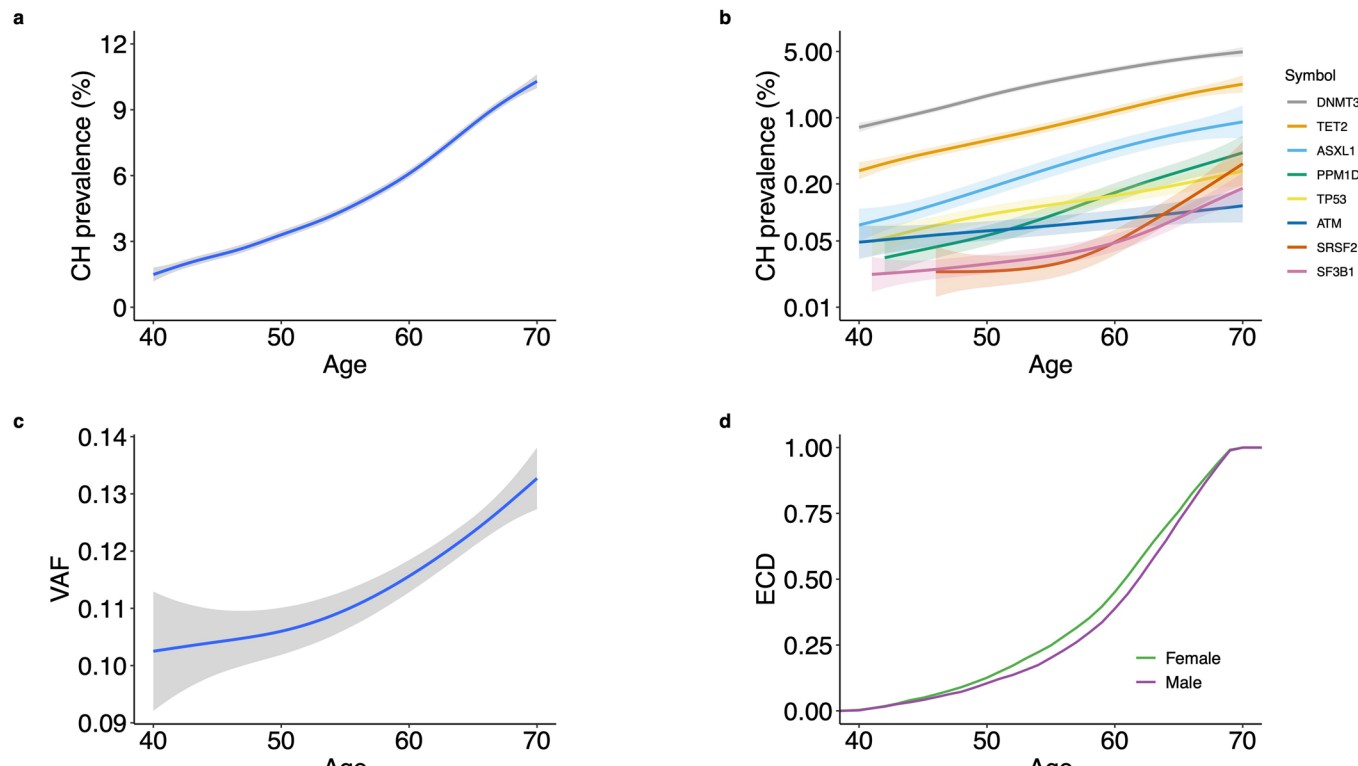

**Extended Data Fig. 2 | Age distribution of CH by mutant gene, clone size, and sex. a**, Prevalence of CH in the cohort with advancing age. The blue line represents the smoothed model fitted to a generalized additive model with 95% confidence interval (CI; gray shadow). **b**, Prevalence of CH by age stratified by the top eight most frequently mutated genes. Colored lines represent the smoothed model fitted to a generalized additive model with 95% CI (colored shadows). Y-axis is log-scaled. **c**, Clone size, estimated by the variant allele fraction (VAF), increases with age. The blue line represents the smoothed model fitted to a generalized additive model and the shadow represents the 95% CI. **d**, Empirical cumulative distribution (ECD) of the age of individuals with CH stratified by sex. CH was observed one year earlier in females than in males (median 61 versus 62 years; $P$=1.6x10$^{-4}$, two-sided pairwise Wilcoxon rank sum test).

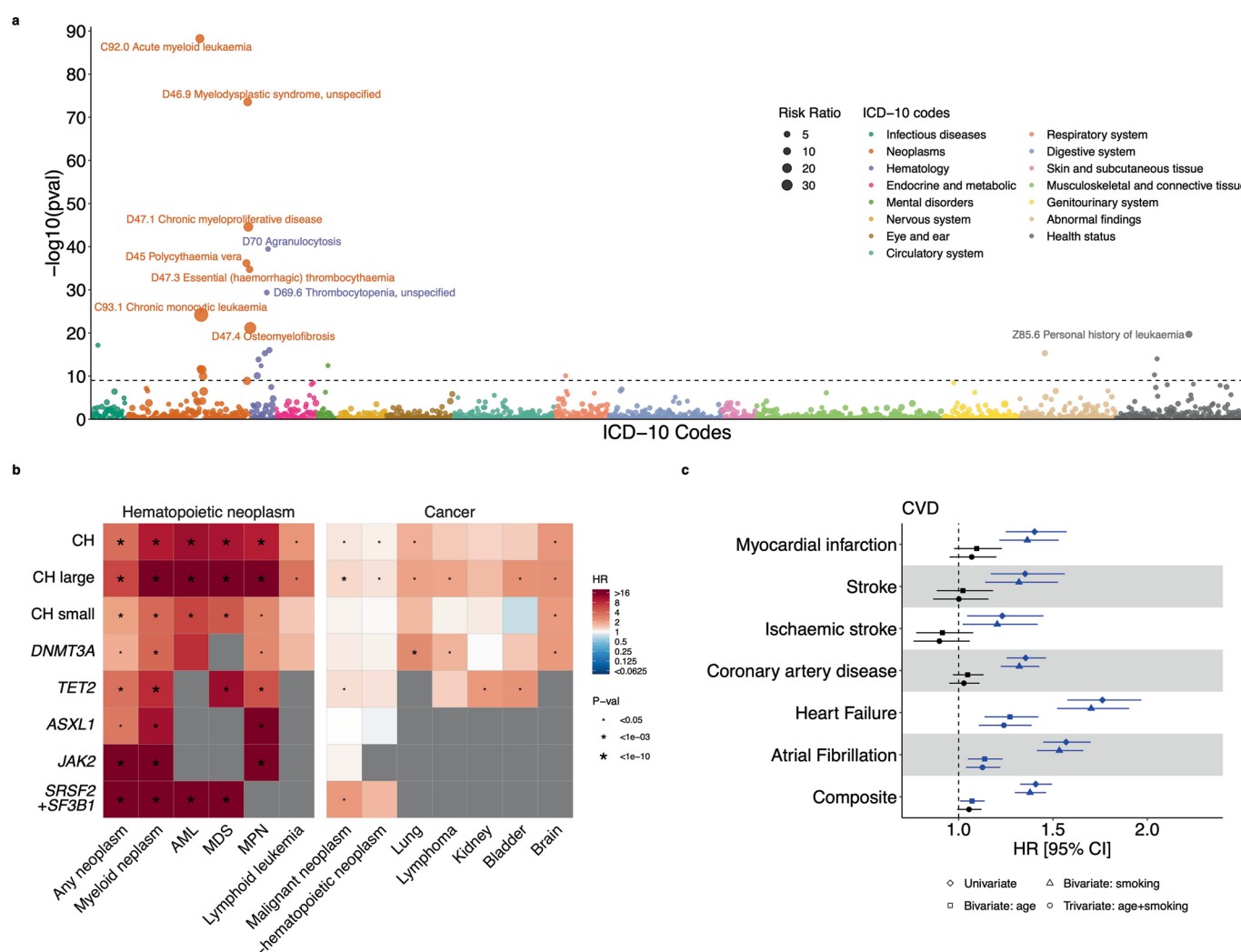

**Extended Data Fig. 3 | Associations between CH and diseases. a**, Phenome-wide association study of CH and incident disease outcomes. Phenotypes were extracted from the International Classification of Diseases version-10 (ICD-10) disease codes and grouped in different categories. A total of 11,787 ICD-10 codes were tested using logistic regression, obtaining results for 2,378. Risk ratio (RR) of each code is represented by a single point with a size scale. The black dashed line represents the phenome-wide significant *P*-value threshold of $10^{-9}$. Only ICD-10 codes with false discovery rate (FDR)<$10^{-15}$ are annotated to control for multiple comparisons. Full results with RRs, 95% confidence intervals (CIs), *P*-values and FDRs are reported in Supplementary Table 8. **b**, Heatmaps showing associations of overall CH (CH), CH with large (CH large) and small (CH small) clones, and CH driven by *DNMT3A*, *TET2*, *ASXL1*, *JAK2*, and *SRSF2+SF3B1* mutations with incident hematopoietic neoplasms and cancer in self-reported non-smokers. Red-blue color scale represents the hazard ratio (HR). HRs were calculated using competing risks models. Gray color represents failure of the logistic regression model (maximum likelihood estimation algorithm) to converge. Asterisk represents a significant association, and its size represents different unadjusted *P*-value cut-offs. All HRs, 95% CIs, sample sizes, *P*-values, and FDRs (to adjust for multiple comparisons) are reported in Supplementary Table 11. **c**, Forest plot showing the hazard ratios (HRs) for cardiovascular disease (CVD) from competing risks analysis in CH using four models: univariate with CH as the only predictor, bivariate including CH and smoking status, bivariate with CH and age, and trivariate with CH, smoking status and age. HR markers with unadjusted *P*<0.05 are depicted in blue. Symbols represent the HRs and error bars represent 95% CIs. All HRs, 95% CIs, sample sizes, *P*-values, and FDRs (to adjust for multiple comparisons) are reported in Supplementary Table 12. Abbreviations: AML, acute myeloid leukemia; MDS, myelodysplastic syndromes; MPN, myeloproliferative neoplasms; CMML, chronic myelomonocytic leukemia; CVD, cardiovascular disease.

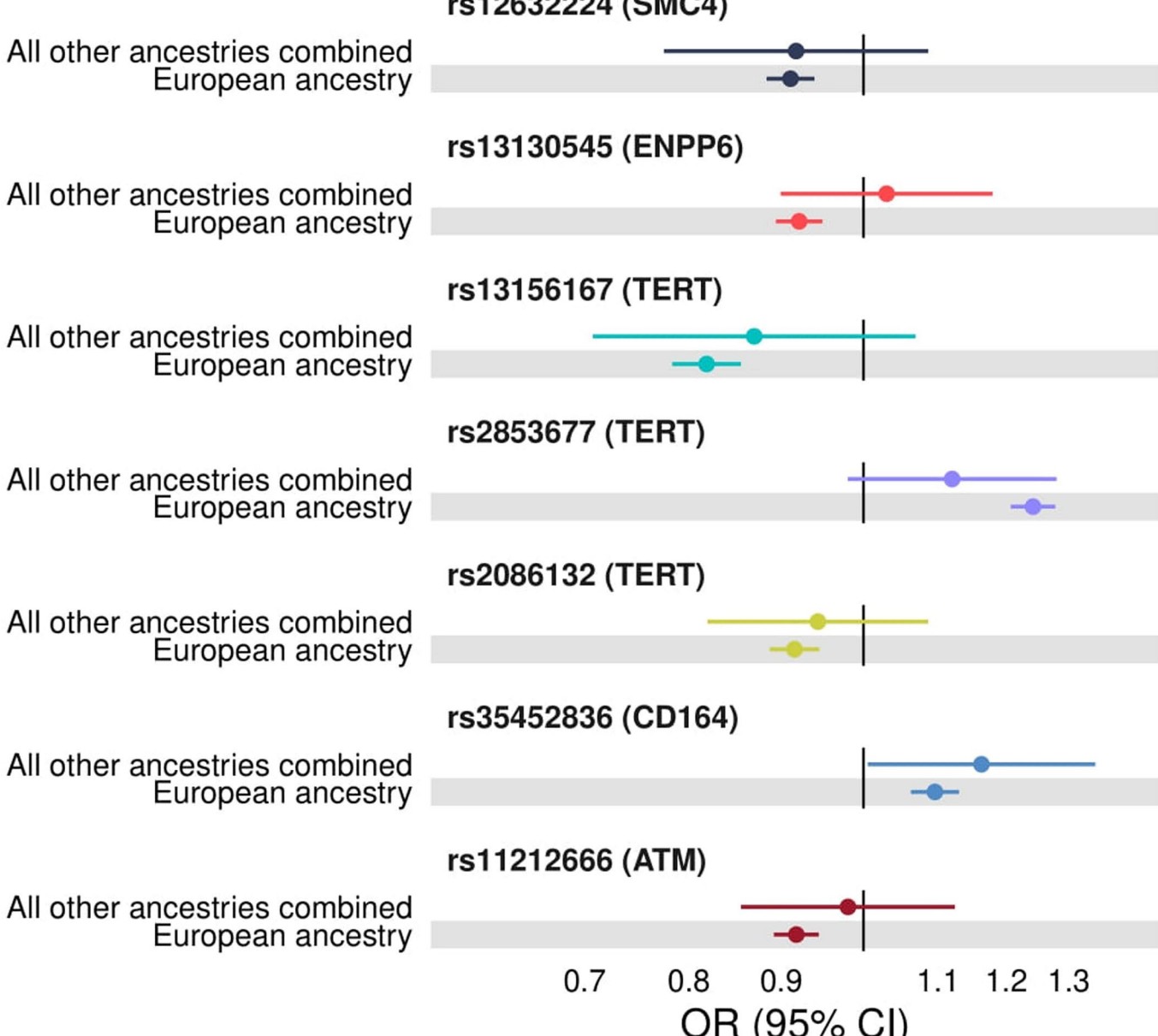

**Extended Data Fig. 4 | Multi-ancestry associations for the seven lead variants for overall CH risk.** Comparison of effect size estimates (odds ratios (ORs)) for the seven overall CH risk lead variants between (i) the 505 individuals with CH and 11,893 controls comprising the ancestrally diverse 'All other ancestries combined' sub-cohort; based on logistic regression and (ii) the 10,203 individuals with CH and 173,918 controls comprising the 'European ancestry' sub-cohort of the 200k UK Biobank cohort; based on linear mixed models (BOLT-LMM). ORs are presented with alignment to the same allele in both sub-cohorts. Symbols represent ORs and error bars represent 95% confidence intervals (CIs).

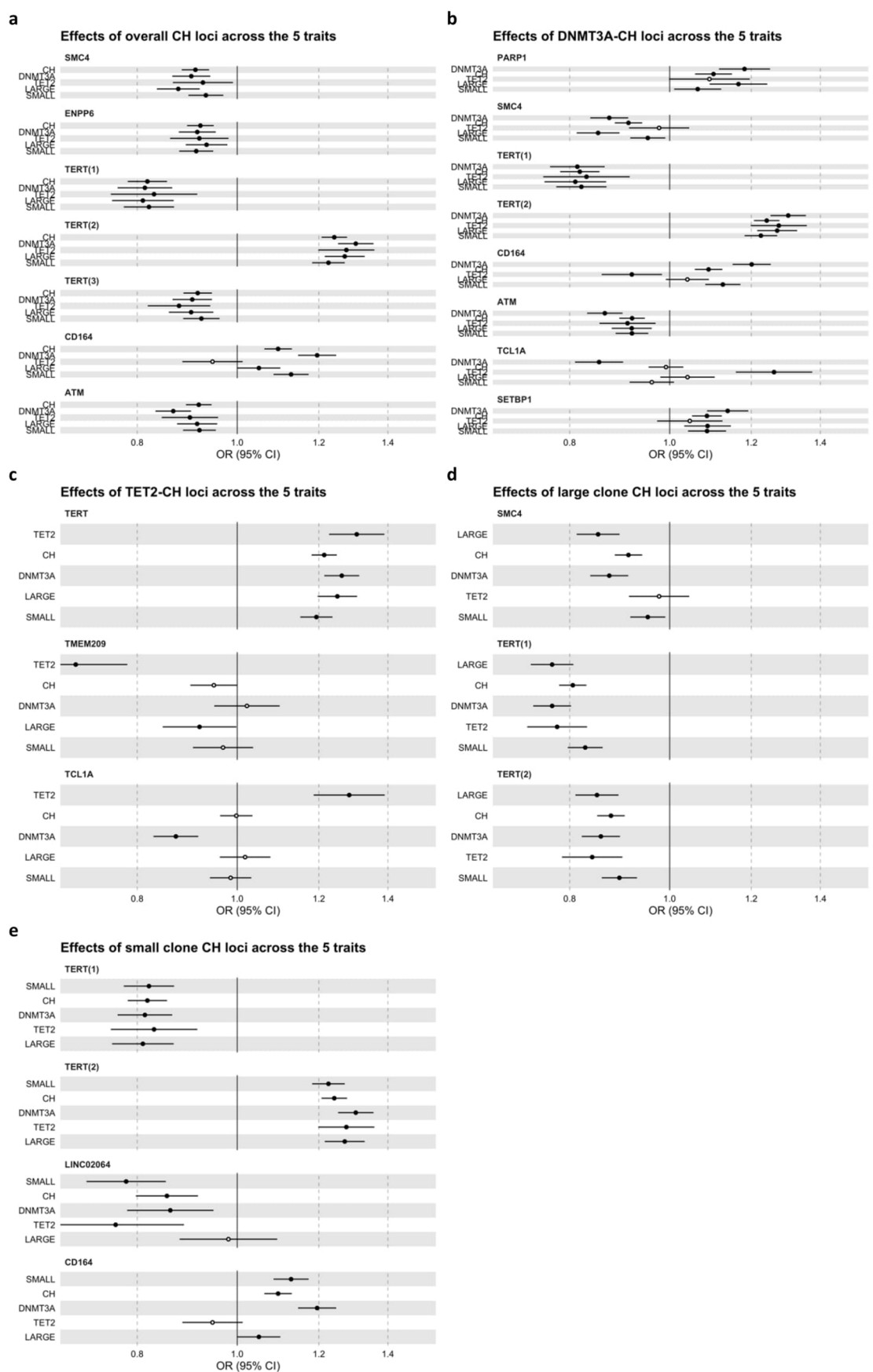

**Extended Data Fig. 5 | See next page for caption.**

**Extended Data Fig. 5 | Heterogeneity of lead GWAS variants across five CH traits.** Forest plots with linear mixed model (BOLT-LMM) odds ratios (ORs) and 95% confidence intervals (CIs) based on data from Supplementary Tables **a**, 16, **b**, 18, **c**, 19, **d**, 20, and **e**, 21. Results for lead variants identified at genome-wide significance ($P<5$x$10^{-8}$) for each CH trait (**a**, overall CH, **b** *DNMT3A*-CH, **c** *TET2*-CH, **d** large clone CH, and **e**, small clone CH) are plotted alongside results for the same lead variants in the four other genome-wide association analyses conducted. Symbols represent ORs and error bars represent 95% confidence intervals (CIs) in **a**, **b**, **c**, **d**, and **e**. Sample sizes: 10,203 individuals with CH ('cases') and 173,918 individuals without CH ('controls') for the overall CH analysis; 5,185 cases and 173,918 controls for *DNMT3A*-CH; 2,041 cases and 173,918 controls for *TET2*-CH; 4,049 cases and 173,918 controls for large clone CH; and 6,154 cases and 173,918 controls for small clone CH.

George S. Vassiliou
Siddhartha P. Kar

# Reporting Summary

## Statistics

For all statistical analyses, confirm that the following items are present in the figure legend, table legend, main text, or Methods section.

| n/a | Confirmed | |
|---|---|---|
| ☐ | ☒ | The exact sample size (*n*) for each experimental group/condition, given as a discrete number and unit of measurement |
| ☐ | ☒ | A statement on whether measurements were taken from distinct samples or whether the same sample was measured repeatedly |
| ☐ | ☒ | The statistical test(s) used AND whether they are one- or two-sided<br>*Only common tests should be described solely by name; describe more complex techniques in the Methods section.* |
| ☐ | ☒ | A description of all covariates tested |
| ☐ | ☒ | A description of any assumptions or corrections, such as tests of normality and adjustment for multiple comparisons |
| ☐ | ☒ | A full description of the statistical parameters including central tendency (e.g. means) or other basic estimates (e.g. regression coefficient) AND variation (e.g. standard deviation) or associated estimates of uncertainty (e.g. confidence intervals) |
| ☐ | ☒ | For null hypothesis testing, the test statistic (e.g. *F*, *t*, *r*) with confidence intervals, effect sizes, degrees of freedom and *P* value noted<br>*Give P values as exact values whenever suitable.* |
| ☒ | ☐ | For Bayesian analysis, information on the choice of priors and Markov chain Monte Carlo settings |
| ☒ | ☐ | For hierarchical and complex designs, identification of the appropriate level for tests and full reporting of outcomes |
| ☐ | ☒ | Estimates of effect sizes (e.g. Cohen's *d*, Pearson's *r*), indicating how they were calculated |

*Our web collection on statistics for biologists contains articles on many of the points above.*

## Software and code

Policy information about availability of computer code

| Data collection | Data were collected by the UK Biobank resource. We downloaded the data from the UK Biobank repository using helper programs (ukb_conv, ukbfetch, ukbgene, ukb_md5, and ukb_unpack) for the whole exome sequencing and phenotype data. |
|---|---|

| Data analysis | We used publicly available software and web-based tools for analyses: somatic variant calling was performed using Mutect2 GATK v4.1.8.1 for the discovery whole exome sequences and using Mutect2 GATK v4.2.2.0 for the replication whole exome sequences (https://github.com/broadinstitute/gatk/releases). Genome-wide association analyses were conducted using BOLT-LMM v2.3.6 (https://alkesgroup.broadinstitute.org/BOLT-LMM/BOLT-LMM_manual.html). LDSC v1.0.1 (https://github.com/bulik/ldsc) was used for heritability estimation and cell type-specific enrichment analyses. Genetic correlations were computed using HDL v1.4.0 (https://github.com/zhenin/HDL). LD-clumping was undertaken using FUMA v1.3.6b (https://fuma.ctglab.nl). Conditional analysis was carried out using the GCTA-COJO tool (https://yanglab.westlake.edu.cn/software/gcta/#COJO) in GCTA v1.93 (https://yanglab.westlake.edu.cn/software/gcta/#Download). Gene-level association analyses were performed using the MAGMA v1.08 tool (https://ctg.cncr.nl/software/magma) in FUMA v1.3.6b (https://fuma.ctglab.nl). Transcriptome-wide association analyses were conducted using SMR v1.03 (https://yanglab.westlake.edu.cn/software/smr/#Download). Protein-protein interaction network and pathway analyses were performed using NetworkAnalyst v3.0 (https://www.networkanalyst.ca). Fine-mapping was conducted using PICS2 v2.1.1 (https://pics2.ucsf.edu). Fine-mapped variant-gene body overlap and fine-mapped variant ATAC-seq peak overlap and ATAC-RNA count correlations were performed using the scripts at https://github.com/sankaranlab/mpn-gwas. SIFT and PolyPhen scores were obtained via the SNPnexus v4 (https://snp-nexus.org). Lead variants were searched on the Open Targets Genetics (https://genetics.opentargets.org) and PhenoScanner V2 (http://www.phenoscanner.medschl.cam.ac.uk) platforms. Druggability of prioritized functional target gene products was evaluated using the Open Targets Platform (https://platform.opentargets.org) and canSAR v1.5.0 (https://cansarblack.icr.ac.uk). For the observational analyses, we used the glm and p.adjust (in the R stats package v4.0.2), coxph (in the R survival package v3.2-11) and crr (in the R cmprsk package v2.2-10) functions, all implemented in R v4.0.2. For the Mendelian randomization analyses, we used the TwoSampleMR v0.5.6 R package implemented in R v4.0.5. Custom code was written for variant calling (and made available as a Nextflow v20.07.1 pipeline) and Mendelian randomization and these scripts are available at https://doi.org/10.5281/zenodo.6419042 (https://github.com/siddhartha-kar/clonal-hematopoiesis). |
|---|---|

For manuscripts utilizing custom algorithms or software that are central to the research but not yet described in published literature, software must be made available to editors and reviewers. We strongly encourage code deposition in a community repository (e.g. GitHub). See the Nature Portfolio guidelines for submitting code & software for further information.

# Data

Policy information about availability of data

All manuscripts must include a data availability statement. This statement should provide the following information, where applicable:
- Accession codes, unique identifiers, or web links for publicly available datasets
- A description of any restrictions on data availability
- For clinical datasets or third party data, please ensure that the statement adheres to our policy

Summary statistics for the overall and subtype-specific CH genome-wide association analyses reported here have been made publicly available at https://doi.org/10.5281/zenodo.5893861. They can also be downloaded from the GWAS Catalog (https://ftp.ebi.ac.uk/pub/databases/gwas/summary_statistics/GCST90102001-GCST90103000/) using the study accession numbers GCST90102618 (overall CH), GCST90102619 (DNMT3A-CH), GCST90102620 (TET2-CH), GCST90102621 (small clone CH), and GCST90102622 (large clone CH). Individual-level UK Biobank data can be requested via application to the UK Biobank (https://www.ukbiobank.ac.uk). The CH call set has been returned to the UK Biobank to enable individual-level data linkage for approved UK Biobank applications. Pre-computed 1000 Genomes phase 3 European ancestry reference panel and cell-type group LD scores used for heritability estimation and cell-type group partitioned heritability analysis, respectively, can be downloaded from https://alkesgroup.broadinstitute.org/LDSCORE. The STRING protein-protein interaction and Reactome pathway databases are available from, and were accessed via, the NetworkAnalyst 3.0 platform (https://www.networkanalyst.ca) and are also downloadable separately from http://version10.string-db.org/cgi/download.pl for STRING v10 and https://reactome.org/download-data for Reactome. Cis-expression quantitative trait locus data from the eQTLGen consortium used for the SMR analyses can be downloaded from https://www.eqtlgen.org/cis-eqtls.html. All summary genetic association statistics data sets used in the Mendelian randomization (MR) and MR-phenome-wide association analyses are publicly available at the Integrative Epidemiology Unit (IEU) OpenGWAS project portal (https://gwas.mrcieu.ac.uk/datasets) and can be accessed by entering the identifiers provided in the "id/link" column in Supplementary Table 35 and the "id.outcome" column in Supplementary Tables 41 and 42 in the "GWAS ID" field at https://gwas.mrcieu.ac.uk/datasets.

# Field-specific reporting

Please select the one below that is the best fit for your research. If you are not sure, read the appropriate sections before making your selection.

☒ Life sciences ☐ Behavioural & social sciences ☐ Ecological, evolutionary & environmental sciences

For a reference copy of the document with all sections, see nature.com/documents/nr-reporting-summary-flat.pdf

# Life sciences study design

All studies must disclose on these points even when the disclosure is negative.

| Sample size | We used all participants (n = 200,453; age range: 38-72, 55% females) in the UK Biobank for whom whole exome sequencing data have been released in December 2020 for the primary analyses. Replication genetic association analyses were based on an additional 221,285 European-ancestry individuals (age range: 39-73, 53% females) in the UK Biobank for whom whole exome sequencing was performed after our UK Biobank discovery set. No statistical method was used to predetermine sample size. |
|---|---|
| Data exclusions | For the discovery genome-wide association analyses, we excluded participants with genetic sex mismatches, participants having non-European ancestries (by self-report or inferred by genetics) or excess heterozygosity (>3 standard deviations from the mean), and included only one of each set of related participants (third-degree relatives or closer). We also performed a targeted association analysis focused on the seven genome-wide significant (P<5e-8) lead variants associated with CH in European-ancestry individuals, evaluating their associations in 505 individuals with CH and 11,893 controls without CH who had diverse (non-European) ancestry and were excluded from the initial discovery genome-wide association analyses. Details are provided in the Methods section of the manuscript. The exclusion criteria were not pre-specified. |

| Replication | Replication was undertaken using independent somatic mutation calling and germline association analysis pipelines on data from 221,285 European-ancestry individuals in the UK Biobank, for whom whole exome sequencing was performed after our UK Biobank discovery set. We focused on DNMT3A and/or TET2 mutation carriers (n=9,386) in the replication sample, stratified by these two genes and clone size, and evaluated the 20 unique lead variants that we identified in the discovery GWAS (representing 26 distinct lead variants for overall/subtype-specific CH associations). Eighteen of 20 variants were replicated at P<0.05, with 16 of the 20 replicating at the Bonferroni threshold of P<0.05/20 and 19 of 20 showing directionally consistent effect size point estimates (Supplementary Table 22). The overall CH variant rs13130545 (4q35.1-ENPP6) and the small clone-specific variant rs72755524 (5p13.3-LINC02064) did not show an association at P<0.05 in the replication analysis. Notably, we confirmed our observation from the discovery GWAS that lead variant alleles at the TCL1A and CD164 loci also had opposite effects on DNMT3A- and TET2-CH in the replication set, and we also replicated the CHEK2 rare variant association.<br><br>Further, the following results also attest to the quality of our data and the validity of our genetic discoveries: replication at P<5e-8 of all previously reported CH risk loci identified in populations of European ancestry at genome-wide significance (P<5e-8), consistency of genome-wide SNP heritability estimates (3.6%) with previously reported estimates of narrow-sense (additive) heritability for CH, demonstration of enrichment of the genome-wide SNP association signal from our analysis in epigenetic marks specific to the hematopietic system, correlation between genome-wide association data for susceptibility to CH and mosaic chromosomal alterations (mCAs; a trait related to CH), including the fact that 13 out of the 19 CH lead variants (P<5e-8) identified by us were associated at P<1e-4 with risk of mCAs in a previously reported GWAS of mCAs (all details, including references, are in the Results section of the manuscript). |
| --- | --- |
| Randomization | The experiments were not randomized. Age at baseline, sex, whole-exome sequencing batch, and first 10 genetic principal components were adjusted for as covariates in BOLT-LMM models for genome-wide association analyses. For observational epidemiological analyses, regression and survival models were adjusted for age, sex, smoking status, whole-exome sequencing batch, and the first ten genetic principal components. In addition, models involving blood cell counts and traits were also adjusted for assessment center while those involving cholesterol and cholesterol species were also adjusted for cholesterol-lowering medication use. |
| Blinding | The experiments were not randomized and investigators were not blinded during the experiments and outcome assessment. The analyses presented in this manuscript do not include any experimental intervention or clinical trial. The investigators involved were not blind to clonal hematopoiesis "case" and control status since genetic association analyses require knowledge of this status. |

# Reporting for specific materials, systems and methods

We require information from authors about some types of materials, experimental systems and methods used in many studies. Here, indicate whether each material, system or method listed is relevant to your study. If you are not sure if a list item applies to your research, read the appropriate section before selecting a response.

## Materials & experimental systems

| n/a | Involved in the study |
| --- | --- |
| ☒ | ☐ Antibodies |
| ☒ | ☐ Eukaryotic cell lines |
| ☒ | ☐ Palaeontology and archaeology |
| ☒ | ☐ Animals and other organisms |
| ☐ | ☒ Human research participants |
| ☒ | ☐ Clinical data |
| ☒ | ☐ Dual use research of concern |

## Methods

| n/a | Involved in the study |
| --- | --- |
| ☒ | ☐ ChIP-seq |
| ☒ | ☐ Flow cytometry |
| ☒ | ☐ MRI-based neuroimaging |

# Human research participants

Policy information about studies involving human research participants

| Population characteristics | The UK Biobank is a prospective longitudinal study containing in-depth genetic and health information from half a million UK participants. The primary analyses in this study were based on all participants (n = 200,453; age range: 38-72, 55% females) in the UK Biobank for whom whole exome sequencing data have been released in December 2020. Replication genetic association analyses were based on an additional 221,285 European-ancestry individuals (age range: 39-73, 53% females) in the UK Biobank for whom whole exome sequencing was performed after our UK Biobank discovery set. |
| --- | --- |
| Recruitment | As stated above, the UK Biobank is a prospective longitudinal study containing in-depth genetic and health information from half a million UK participants. Details of UK Biobank participant recruitment are available at: https://www.ukbiobank.ac.uk and from Sudlow C, et al. (2015) PLoS Med 12(3): e1001779. For this study, we have selected 200,453 individuals who had whole-exome sequencing (WES) data released as of December 2020. Notably, participants were not selected in any way, however, as is the case for several such cohorts, there is some evidence of selection bias in favor of healthier, older, female, and socio-economically better off volunteers (Fry A, et al. Am. J. of Epidemiol., Vol. 186, Issue 9, 1 Nov. 2017, Pgs. 1026–1034.). Also, despite a relatively low response rate to invitations to participate, it has been shown that risk factor associations identified in the UK Biobank are generalizable (Batty GD, et al. BMJ 2020; 368:m131). |
| Ethics oversight | The UK Biobank resource was approved by the North West Multi-centre Research Ethics Committee under reference number 21/NW/0157 and all participants provided written informed consent to participate. Participants in the UK Biobank resource are volunteers and not compensated for participation. Data from the UK Biobank resource were accessed under approved application numbers 56844, 29202, and 26041 for this study. Further details can be found at: https://www.ukbiobank.ac.uk/learn-more-about-uk-biobank/about-us/ethics. |

Note that full information on the approval of the study protocol must also be provided in the manuscript.

