## [Peer Review File · Nature Genetics]

Peer Review Information

Manuscript Title: Genome-wide analyses of 200,453 individuals yield new insights into the causes and consequences of clonal hematopoiesis

Corresponding author name(s): Prof George Vassiliou

Reviewer Comments & Decisions:

Decision Letter, initial version:
--

16th Nov 2021

Dear Professor Vassiliou,

Your Article, "Genome-wide analyses of 200,453 individuals yields new insights into the causes and consequences of clonal hematopoiesis" has now been seen by 3 referees. You will see from their comments below that while they find your work of interest, some important points are raised. We are interested in the possibility of publishing your study in Nature Genetics, but would like to consider your response to these concerns in the form of a revised manuscript before we make a final decision on publication.

To guide the scope of the revisions, the editors discuss the referee reports in detail within the team, including with the chief editor, with a view to identifying key priorities that should be addressed in revision and sometimes overruling referee requests that are deemed beyond the scope of the current study. In discussing these review reports, we were pleased that the reviewers broadly consider the work to be technically sound. With that said, there are a few technical queries which must be addressed (including the issue of replication). We think that many of the reviewer comments have been designed to draw out the novelty of the work, and showcase how it is distinctive from other studies in this space. As such, you're encouraged to take on board suggestions regarding how you discuss your findings in the light of other (often conflicting) papers, and cite past papers. This will - we believe - help place your study in its appropriate scientific context. We agree that the overall impact of the work will be further boosted by making textual changes to improve clarity and focus where suggested. We also note that reviewers have queried data availability and we would like to remind you that the deposition of summary statistics must be made publicly available prior to acceptance. Please could you include an update on your data availability plans in your cover letter.

We hope that you will find the prioritized set of referee points to be useful when revising your study. Please do not hesitate to get in touch if you would like to discuss these issues further.

We therefore invite you to revise your manuscript taking into account all reviewer and editor comments. Please highlight all changes in the manuscript text file. At this stage we will need you to upload a copy of the manuscript in MS Word .docx or similar editable format.

*2) If you have not done so already please begin to revise your manuscript so that it conforms to our Article format instructions, available [here](http://www.nature.com/ng/authors/article_types/index.html). Refer also to any guidelines provided in this letter.

[REDACTED]

Nature Genetics is committed to improving transparency in authorship. As part of our efforts in this direction, we are now requesting that all authors identified as 'corresponding author' on published papers create and link their Open Researcher and Contributor Identifier (ORCID) with their account on the Manuscript Tracking System (MTS), prior to acceptance. ORCID helps the scientific community achieve unambiguous attribution of all scholarly contributions. You can create and link your ORCID from the home page of the MTS by clicking on 'Modify my Springer Nature account'. For more

information please visit please visit www.springernature.com/orcid.

Sincerely,

Safia Danovi
Editor
Nature Genetics

Referee expertise:

Referee #1: GWAS/TWAS

Referee #2: GWAS

Referee #3: CH

Reviewers' Comments:

Reviewer #1:

Remarks to the Author:

Summary: CH was called from exome data in the UKBiobank, identifying ~10k/~200k carriers. CH was associated with age, blood values, and smoking at baseline; as well as with incident blood cancers and cancer-related traits after baseline. Surprisingly, no association was observed between GWAS was carried out and a heritability enrichment analysis identified striking enrichment for enhancers in hematopoietic cells. The GWAS identified 7 significant loci including multiple independent associations in the TERT region, and prioritized putative genes.

Overall, this is a comprehensive analysis of an important phenotype, which identifies novel associations and carries out interesting mendelian randomization studies. The data, methodology, and statistical analyses are sound and the conclusions are generally appropriate. My primary concerns are regarding the lack of replication and the discrepancy with previous findings related to CAD/stroke.

Major Comments:

* Lack of replication data tempers some of the impact of the novel GWAS associations. However this is addressable now that the full set of UKBiobank exomes has been released to investigators. This data should be incorporated into the analysis (at least for GWAS discovery/replication) as using the interim release leaves the current study somewhat half-baked and quickly superseded.

* 412: "These associations taken together with the fact that age and smoking are strong risk factors for CH raise the possibility that previously reported associations of CH with CAD and stroke risks may

suffer from residual confounding.": This is potentially an extremely important finding given that it contradicts the key results of Jaiswal et al. 2017 NEJM, that CH carriers have a 2x higher CAD risk and 4x higher MI risk, which was also observed for individual gene mutations and unlikely to be due to small sample size. If it is the case that previous findings were due to residual confounding from age or smoking, it should be possible to demonstrate this confounding in the UKBiobank data by ascertaining CAD cases/controls or by dropping age/smoking as covariates. This is such a fundamental discrepancy it's critical to understand whether the confounding is with Jaiswal et al. or in this study.

* Given the association of CH with smoking, are some of the associations of CH with incident disease actually driven by smoking rather than CH? It looks like this analysis was investigated for lung cancer (Extended Fig 3d) but would also make sense for other incident disease associations (by looking in non-smokers or adjusting for smoking).

* For the sub-CH GWAS analysis (lines 190) it would be useful to also estimate the genetic correlation between all pairs of analyzed traits (e.g. using LDSC or, ideally, GCTA with individual-level data) Given the relatively small number of individual associations, genetic correlation will be informative of the genome-wide relationships between these phenotypes.

* Full summary association statistics for all of the new GWAS in this study should be made publicly available upon acceptance. This is the standard for GWAS of the UK Biobank and there is no apparent reason not to do so here.

* Code availability are currently placeholder repositories with no documentation.

Minor Comments:

* 489: "Individuals with myeloid malignancies or hematological neoplasms at baseline were excluded from the analysis". Please clarify how myeloid malignancies or hematological neoplasms at baseline were determined.

* 492: "The cancer/CVD/death event was used as an outcome and CH was considered as the exposure in these analyses". For the non-death outcomes how was the competing risk of death addressed? Were proportional hazard assumptions met (it would also be helpful to include Kaplan-Meier plots for the significant results)? What was the start point for this analysis? Were any patients censored and, if so, for what criteria?

* 639: Pleiotropy-robust sensitivity analyses are mentioned but results were not presented, was there evidence of pleiotropy?

Reviewer #2:

Remarks to the Author:

Authors conducted a genome-wide association study (GWAS) of clonal hematopoiesis (CH) using the UK biobank resource of >200k individuals. By assessing the somatic mutations in the ~40 genes responsible for CH, they identified >10k individuals with CH. CH was characterized by age and female/male ratio, as reported previously. Clinical association of CH with traits and diseases observed links with smoking, heart failure (but not apparent for MI), hematological cancers, and so on. The

GWAS identified ~7 loci with genome-wide significance. Stratified GWAS based on clone sizes provided further associated loci. The GWAS follow-up analysis prioritized functional genes (e.g., CD164) and depicted gene-gene networks. Mendelian randomization (MR) analysis handling CH as both exposure and outcome phenotypes identified causal inference. This is a well-designed and highly-powered genetic studies comprehensively assessing genetic backgrounds of CH. This reviewer has few comments.

1. Since this is a GWAS study using a public resource, public deposit of the full GWAS summary statistics is necessary.
2. The current GWAS does not seem to assess the X chromosome. How about an X chromosome-wide association study?
3. Stratified GWAS results of the CH clone sizes should imply interaction/quantitative associations between CH clone size, CH status, and genotypes. Can the authors assess such associations for the identified loci?
4. As for the clinical associations between CH status and diseases/traits, this reviewer recommends to assess wider ranges of phenotypes in a phenome-wide manner, since they are using the UK biobank resource. This may also true for the MR analysis.
5. Regarding the prioritized functional genes, enrichment analysis with target drugs is interesting. Is there enrichment with the targets of the diseases with altered comorbidity with CH (e.g., heart failure and neoplasm)?
6. How about the GWAS results of the non-European UK biobank participants?

Reviewer #3:

Remarks to the Author:

In the manuscript "Genome-wide analysis of 200,453 individuals yields new insights into the causes and consequences of clonal hematopoiesis" Kar et al. use genetic epidemiology approaches to describe novel germline associations with CH and define a driver specificity for these associations that may explain phenotypic differences observed in DNMT3A and TET2 CH for example.

Overall the amount of analysis the authors have done in this work is commendable. However, there are several concerns regarding the manuscript focus and flow that need to be addressed. There are a very large number of observations reported, but they are not always related or well discussed. A large portion of the findings have been reported previously in other datasets. Some are novel, some p-values that are significant in prior datasets are not significant here, and some p-values are significant here that were not significant in prior datasets. I wonder whether the cleanest paper is to focus more on the germline predisposition findings, which are clearly better powered in this larger dataset compared to previous studies.

1. The manuscript reads as multiple separable efforts that include the phenotype of CH, CH pathogenesis, germline associations. The bulk of the analysis focuses on CH pathogenesis and

germline associations. However the authors begin the paper describing (on pages 2-3) phenotypic associations (prevalent and incident diseases) and it is extremely challenging for the reader to relate these findings with those of the germline associations. The discussion on how identified associations may be mechanistically linked to the outcomes described is minimal. Some of the associations observed have confidence intervals that approach 1 (For example the association of BMI has a modest p-value and 95% CI of 1.01-1.31) which call into question the real-world relevance of the statistical association. Though the experimental approach is observational, the work would be significantly strengthened by a discussion of potential mechanism that links important associations. Without this discussion, the volume of associations made in this manuscript without a clear focus does more to convolute understanding of CH consequences.

2. Similarly, the figures do not highlight the novel findings or relate well to each other. Figure 1 is very consistent with prior reported findings with minimal novelty. Figure 2 has many findings that have been reported before and some novel ones, but they are buried. Figure 4 is the most clear figure in terms of what is known and novel.

3. The significance of the ATAC-Seq data and the implications of these findings are not entirely clear to me.

4. Are p-values adjusted for multiple hypotheses testing? This is particularly important in assessing the relevance of mild-modest effect estimates with mild-modest p-values.

5. With regards to the phenotype data, in the section "associations between CH and incident disease" on page 3, the authors report that CH is not associated with ischemic cardiovascular disease in the UKB. They do not provide any discussion of this novel finding – are there differences in cohort composition for example that might explain this?

6. On page 4, the authors describe associations of germline TERT variants and long leukocyte telomere length with CH. Within the discussion, this finding should be reconciled this with recent data from Reilley et al Blood 2021 (doi: 10.1182/blood.2021011075) which identifies germline variants and shortened telomeres as being associated with myelodysplastic syndrome.

7. On page 5 the authors discuss the hematological association of autosomal mCA with CH. The relationship between CH and autosomal mCA in both myeloid and lymphoid CH has recently been described using data from the UK Biobank. The authors should reference this in their discussion. <https://doi.org/10.1038/s41591-021-01521-4>

Minor suggestions:

- In abstract on page 1, line 36 and on page 8, line 363 – would change text to "CH from 4 to 14" for uniformity
- The text within Figure 4 is difficult to read within the boxes given the chosen colors
- In extended figure 3b are myeloid neoplasms a composite of several ICD10 codes or derived from codes shown in figure 3a. This needs to be specified in the methods. "Thrombocytopenia unspecified" should not be used as this is a basket code for many different causes of thrombocytopenia, most non-malignant.

Author Rebuttal to Initial comments

We thank our Reviewers for their positive overall assessment of our manuscript and for their genuinely helpful and insightful comments. We have now addressed these by performing new analyses, generating new figures, and making relevant changes to the text that help improve clarity and emphasise the most salient of our novel findings. We believe that, as a result of these changes/edits, our manuscript has been substantially strengthened and represents a comprehensive and authoritative study of the causes and consequences of the common and important phenomenon of clonal hematopoiesis.

Below, we provide a point-by-point response to their comments and refer our Reviewers to the relevant changes/additions to our manuscript. Reviewer comments are in black, our responses in dark blue and the actions we have undertaken for the revision in dark red. In our revised manuscript, changes/additions made in response to Reviewers' comments are also in dark red.

Reviewers' Comments:

Reviewer #1:

Remarks to the Author:

Summary: CH was called from exome data in the UKBiobank, identifying ~10k/~200k carriers. CH was associated with age, blood values, and smoking at baseline; as well as with incident blood cancers and cancer-related traits after baseline. Surprisingly, no association was observed between GWAS was carried out and a heritability enrichment analysis identified striking enrichment for enhancers in hematopoietic cells. The GWAS identified 7 significant loci including multiple independent associations in the TERT region, and prioritized putative genes.

Overall, this is a comprehensive analysis of an important phenotype, which identifies novel associations and carries out interesting mendelian randomization studies. The data, methodology, and statistical analyses are sound and the conclusions are generally appropriate. My primary concerns are regarding the lack of replication and the discrepancy with previous findings related to CAD/stroke.

We thank Reviewer #1 for their positive overall assessment of our work and recognise their two main concerns pertaining to replication and discrepancy with previous findings relating to CAD/stroke. As discussed below, we respond to both important concerns in what we hope is a detailed and persuasive manner.

Major Comments:

* Lack of replication data tempers some of the impact of the novel GWAS associations. However, this is addressable now that the full set of UKBiobank exomes has been released to investigators. This data should be incorporated into the analysis (at least for GWAS discovery/replication) as using the interim release leaves the current study somewhat half-baked and quickly superseded.

Acting on the Reviewer's suggestion we have now incorporated the additional UK Biobank (UKB) exomes in a discovery-replication GWAS design. Replication was undertaken using an independent somatic mutation calling strategy and germline association analysis pipeline and data from 221,285 European-ancestry individuals in the UKB, for whom whole exome sequencing was performed after our UKB discovery sample set. We focused on *DNMT3A* and/or *TET2* mutation carriers only in the replication set (n = 9,386), stratified by these two

genes and clone size, and evaluated the 20 unique lead variants that we identified in the discovery GWAS (representing 26 distinct lead variant overall or subtype-specific CH associations). Eighteen of the 20 variants were replicated at $P < 0.05$, with 16 of 20 replicating at the Bonferroni threshold of $P < 0.05/20$ and 19 of 20 showing consistent directionality and effect size point estimates in the replication set (full replication results in new Supplementary Table 22). Notably, we confirmed in the replication set our observation from the discovery GWAS that lead variant alleles at the *TCL1A* and *CD164* loci had opposite effects on *DNMT3A*- and *TET2*-CH, and also replicated the *CHEK2* rare variant association. We have added these findings to a new section, "Replication of genome-wide significant lead variants" under Results (page 5) and a corresponding section under Methods (page 12) and also comment on the replication in Discussion (page 8).

* 412: "These associations taken together with the fact that age and smoking are strong risk factors for CH raise the possibility that previously reported associations of CH with CAD and stroke risks may suffer from residual confounding.": This is potentially an extremely important finding given that it contradicts the key results of Jaiswal et al. 2017 NEJM, that CH carriers have a 2x higher CAD risk and 4x higher MI risk, which was also observed for individual gene mutations and unlikely to be due to small sample size. If it is the case that previous findings were due to residual confounding from age or smoking, it should be possible to demonstrate this confounding in the UKBiobank data by ascertaining CAD cases/controls or by dropping age/smoking as covariates. This is such a fundamental discrepancy it's critical to understand whether the confounding is with Jaiswal et al. or in this study.

We are grateful to the Reviewer for this critically important question. A very similar point is also raised by Reviewer #3 (question 5) and we provide below our consolidated response to this point for both our Reviewers.

We now follow the suggestion of Reviewers #1 and #3 to explore the evidence for confounding by age and smoking further and examine the association between CH and coronary artery disease (CAD)/stroke with and without age and/or smoking as covariates. We present results for four models in the figure copied below – univariate (including CH as the only predictor), bivariate with CH + age, bivariate with CH + smoking, and trivariate with CH + age + smoking as predictors. We apply these models to myocardial infarction (MI), CAD, ischemic stroke, overall stroke, heart failure (HF) and atrial fibrillation (AF) as individual outcomes, as well as to a composite of cardiovascular outcomes. As evident in the forest plots below, we find that the associations between CH and CAD/MI and CH and stroke/ischemic stroke attenuate to the null once age is included in the models. While there is also substantial attenuation of the associations between CH and HF and CH and AF in models that include age, the HF and AF associations continue to remain significant. The impact of smoking on the associations in the models is relatively small compared to the profound impact of age as a confounder, although in every case the trivariate models that include both age and smoking are the ones closest to or at the null. We now present these new analyses in Extended Data Fig. 3c and Supplementary Table 12 and comment on this under "Associations between CH and incident disease" in the Results section (page 3) and in the Discussion section (page 9).

We also note that there are other recent publications that support our finding of a lack of an association in the UK Biobank (UKB). Notably, Bhattacharya, et al. (2021; <https://pubmed.ncbi.nlm.nih.gov/34743536/>), published after we first submitted our manuscript, reported equivalent findings for the association between stroke and CH in the UKB. For convenience, we have copied below the relevant figure (Supplementary Fig. S12F) from Bhattacharya, et al., which demonstrates the lack of an association between CH and ischemic stroke amongst ~45,000 UKB participants (a subset of the ~200k analyzed in our study). The confidence interval for the association continues to include the null even after inclusion of multiple other studies, including several studies that are highly enriched for stroke outcomes.

Figure 1. Forest plot of meta-analyzed hazard ratio for the association between clonal hematopoiesis of indeterminate potential (CHIP) and stroke.
 Cox proportional hazards models were fitted, adjusted for age, sex, and the first 10 principal components of genetic ancestry. Here, forest plots are used to show the hazard ratio (HR), 95% CI, and numerical events for each study. **A.** CHIP was associated with a meta-analyzed HR of 1.14 (95% CI, 1.03-1.27) for all incident stroke. **B.** CHIP was associated with a meta-analyzed HR of 1.11 (95% CI, 0.98-1.25) for all incident ischemic stroke. **C.** CHIP was associated with a meta-analyzed HR of 1.24 (95% CI, 1.01-1.51) for all incident hemorrhagic stroke. ARIC indicates Atherosclerosis Risk in Communities; CHS, Cardiovascular Health Study; FHS, Framingham Heart Study; JHS, Jackson Heart Study; MESA, Multi-Ethnic Study of Atherosclerosis; MGBB, Mass General Brigham Biobank; UKBB, United Kingdom Biobank; and WHI, Women's Health Initiative.

It is noteworthy that many of the cohorts that have reported the association are enriched in participants at high cardiovascular risk (e.g., Bioimage, ATVB, and PROMIS in Jaiswal, et al. 2017). One plausible mechanistic explanation for the lack of an association in the UKB is that the UKB cohort, in contrast, is relatively younger and healthier and may therefore potentially have lower rates of epigenetic aging. Recently published findings, also from the Jaiswal group (Nachun, et al. 2021; <https://pubmed.ncbi.nlm.nih.gov/34050697/>), suggest that CH was associated with CAD/stroke only on a background of epigenetic aging, whereas the association was not seen in those without evidence of epigenetic age acceleration. **Based on our Reviewer #3's suggestion to provide an expanded discussion on mechanisms we include the above under the Discussion section (page 9).**

* Given the association of CH with smoking, are some of the associations of CH with incident disease actually driven by smoking rather than CH? It looks like this analysis was investigated for lung cancer (Extended Fig 3d) but would also make sense for other incident disease associations (by looking in non-smokers or adjusting for smoking).

We thank the Reviewer for this suggestion. We would like to note that all associations of CH with incident disease that were presented in the original submission were based on regression models that adjusted for smoking status as a covariate. As the Reviewer points out, we also presented results for lung cancer stratified by smoking status. Now, in line with the Reviewer's suggestion, we have extended this stratified analysis to other incident disease associations and show that the associations of CH with myeloid neoplasms and multiple solid tumor types are also seen in non-smokers, indicating that these associations are independent of smoking status (please see revised Extended Data Fig. 3b, which we also copy here for the Reviewer's convenience – the associations shown in this figure are in non-smokers). We highlight these updated analyses in the Results section (page 3) under "Associations between CH and incident disease".

b

* For the sub-CH GWAS analysis (lines 190) it would be useful to also estimate the genetic correlation between all pairs of analyzed traits (e.g. using LDSC or, ideally, GCTA with

individual-level data) Given the relatively small number of individual associations, genetic correlation will be informative of the genome-wide relationships between these phenotypes.

We are grateful to the Reviewer for this helpful suggestion. In the original submission we did not present genetic correlations for the CH subtypes (correlation between *DNMT3A*-CH and *TET2*-CH and between large clone size CH and small clone CH) because the smaller sample sizes for these subtype collections prevented calculation of genetic correlation via LDSC. However, in the revised submission, we have now calculated these genetic correlations using the more powerful high-definition likelihood (HDL) inference approach (Ning, et al. 2020; <https://pubmed.ncbi.nlm.nih.gov/32601477/>). We find a negative genetic correlation ($r_g=-0.48$, s.e.=0.33) between *DNMT3A*-CH and *TET2*-CH and while this correlation does not reach statistical significance ($P=0.15$), the relatively large negative correlation further supports our statement in the Discussion on page 8 that “This inverse relationship...is tantalizing in light of recent observations that ageing has different effects on the dynamics of these two forms of CH, resulting in *TET2*-CH becoming more prevalent than *DNMT3A*-CH in those aged over 80 years”. We find the genetic correlation between large and small clone CH to be 0.37 (s.e.=0.18, $P=0.018$) indicating that while large and small clone size CH are genetically correlated there are, nonetheless, differences in the underlying genetics of these two CH phenotypes. We report these new findings in the Results section under “CH GWAS stratified by gene and clone size, and association heterogeneity” (page 5) with a corresponding entry in the Methods section under “Linkage disequilibrium score regression (LDSC) and high-definition likelihood (HDL) inference” (page 13) and in the Discussion (page 8).

* Full summary association statistics for all of the new GWAS in this study should be made publicly available upon acceptance. This is the standard for GWAS of the UK Biobank and there is no apparent reason not to do so here.

We have now uploaded the full summary association statistics for all of the new GWAS in this study to <https://zenodo.org/record/5893861> and include this link under the Data Availability section (page 15).

* Code availability are currently placeholder repositories with no documentation.

We apologize for the placeholder and have now updated the Code Availability section and linked repository (page 16).

Minor Comments:

* 489: "Individuals with myeloid malignancies or hematological neoplasms at baseline were excluded from the analysis". Please clarify how myeloid malignancies or hematological neoplasms at baseline were determined.

We excluded from the association analysis all individuals with a cancer diagnosis date prior to the date they attended the assessment centre (baseline). Determination of cancer history was based on the fields: self-reported cancer, self-reported illness, and ICD-10 and ICD-9 codes (please see revised Supplementary Table 9). We have now clarified this point in the Methods section under “Trait selection and modelling for the conventional observational multivariable regression analyses” (page 11).

* 492: "The cancer/CVD/death event was used as an outcome and CH was considered as the exposure in these analyses". For the non-death outcomes how was the competing risk of death addressed? Were proportional hazard assumptions met (it would also be helpful to include Kaplan-Meier plots for the significant results)? What was the start point for this analysis? Were any patients censored and, if so, for what criteria?

In our original submission, we did not address competing risk, but we did censor individuals who died without the event at the time of death. To improve our model following the Reviewer's suggestion, we have now repeated all cancer and CVD analyses using a competing risk model. Of note, results were very similar to the previous Cox proportional hazards model (please see revised Figure 2). The proportional hazards assumption for the Cox and competing risk models was assessed by examining the Schoenfeld residuals and the assumption was met for the results presented. The starting point was established on the date that people attended the assessment centres. The description of this new model, including the start date, has been added to the Methods section under "Trait selection and modelling for the conventional observational multivariable regression analyses" (page 11).

* 639: Pleiotropy-robust sensitivity analyses are mentioned but results were not presented, was there evidence of pleiotropy?

We apologize for not clarifying this in the text in the Results section of the original submission. Results for the two most common pleiotropy-robust Mendelian randomization (MR) sensitivity analysis methods – the weighted median and MR-Egger methods – were presented alongside the primary inverse-variance weighted (IVW) MR method results in Supplementary Tables 35 to 41 (please note the revised supplementary table numbering – we now reference the MR-Egger and weighted median analyses and the tables in the Results section under "Mendelian randomization (MR) to uncover the causes and consequences of CH" (page 8) to guide the reader). For the key MR findings that we highlight in the aforementioned section, the two pleiotropy-robust methods provided effect size estimates that were largely consistent with the primary IVW MR method.

Reviewer #2:

Remarks to the Author:

Authors conducted a genome-wide association study (GWAS) of clonal hematopoiesis (CH) using the UK biobank resource of >200k individuals. By assessing the somatic mutations in the ~40 genes responsible for CH, they identified >10k individuals with CH. CH was characterized by age and female/male ratio, as reported previously. Clinical association of CH with traits and diseases observed links with smoking, heart failure (but not apparent for MI), hematological cancers, and so on. The GWAS identified ~7 loci with genome-wide significance. Stratified GWAS based on clone sizes provided further associated loci. The GWAS follow-up analysis prioritized functional genes (e.g., CD164) and depicted gene-gene networks. Mendelian randomization (MR) analysis handling CH as both exposure and outcome phenotypes identified causal inference. This is a well-designed and highly-powered genetic studies comprehensively assessing genetic backgrounds of CH. This reviewer has few comments.

We thank Reviewer #2 for their positive overall assessment of our work.

1. Since this is a GWAS study using a public resource, public deposit of the full GWAS summary statistics is necessary.

We have now uploaded the full summary statistics for all of the new GWAS in this study to <https://zenodo.org/record/5893861> and include this link under the Data Availability section (page 15).

2. The current GWAS does not seem to assess the X chromosome. How about an X chromosome-wide association study?

We thank our Reviewer for this astute comment and apologize for not including this in the original submission. We have now performed an X chromosome-wide association study and include results for the X chromosome in the full GWAS summary statistics files that we have now made publicly available. In line with our autosomal association analyses, we evaluated variants with minor allele frequency (MAF)>1% and imputation quality score>0.6, yielding 298,048 chromosome X variants, but unfortunately none of these were genome-wide significant ($P<5\times 10^{-8}$). Further, there were no genome-wide significant ($P<10^{-9}$) associations in the MAF 0.2%-1% (imputation quality score>0.8) range either. We have now updated the total number of variants evaluated in the GWAS, which we report on page 3 (under “Germline genetic loci associated with overall CH susceptibility” in the Results section), and Figure 4 (the five Manhattan plots) to include the X chromosome results and note the inclusion of the X chromosome in our analysis in the Methods section under “Germline genotype data processing and genome-wide association analyses” (page 12).

3. Stratified GWAS results of the CH clone sizes should imply interaction/quantitative associations between CH clone size, CH status, and genotypes. Can the authors assess such associations for the identified loci?

We agree with the Reviewer that this is a point of great interest. We presented stratified GWAS results in Extended Data Fig. 5a-c (we copy the figure below for the Reviewer’s convenience) where we examined the lead variants identified for overall, *DNMT3A* and *TET2* CH in the context of large and small clone size CH. We found that, in general, the effect size point estimates for these lead variants for large and small clone size CH were similar in magnitude and direction to the point estimates for overall, *DNMT3A* and *TET2* CH, with three notable exceptions. The lead variants rs138994074 at 1q42.12-*PARP1* and rs12632224 at 3q25.33-*SMC4* had greater effects on large clone than small clone CH while rs35452836 at 6q21-*CD164* had a greater effect on small clone than large clone CH. We now elaborate on these findings in the Results section on under “CH GWAS stratified by gene and clone size, and association heterogeneity” (page 5).

For the lead variants associated at genome-wide significance ($P < 5 \times 10^{-8}$) with *DNMT3A*-CH and/or *TET2*-CH (Supplementary Tables 18 and 19), we also evaluated the association between germline genotype and CH subtype-specific clone size (i.e., between these lead variants and large or small clone *DNMT3A*-CH and large or small clone *TET2*-CH). However, evidence of interaction between genotype, clone size and CH subtype status was not apparent when we visualized these associations (Figure shown below for the Reviewer), although it must be noted that the sample sizes for these analyses were limited compared to the unrestricted subtype and clone size-specific analyses shown in the Figure above.

4. As for the clinical associations between CH status and diseases/traits, this reviewer recommends to assess wider ranges of phenotypes in a phenome-wide manner, since they are using the UK biobank resource. This may also true for the MR analysis.

In the original submission, we reported a phenome-wide association analysis of 11,787 outcomes linked to CH as an exposure by using ICD-10 codes, obtaining results for 2,378 of them (for the remainder, the models (more specifically, the maximum likelihood estimation algorithm) failed to converge likely due to inadequate sample sizes). We apologize for the lack of clarity about this in the original submission. We have now revised Extended Data Fig. 3a to make this clearer, and also clarified this point in the Methods section (page 11) under “Trait selection and modelling for the conventional observational multivariable regression analyses”.

We thank the Reviewer for recommending a Mendelian randomization (MR)-PheWAS as well. We have now taken the genetic instruments for overall CH and *DNMT3A*-CH, since these contained the largest number of variants, and evaluated the associations between genetic predisposition to overall/*DNMT3A*-CH as exposures and 1,434 outcomes (diseases and traits) from the UK Biobank using the TwoSampleMR/IEU OpenGWAS pipeline (mrcieu.github.io/TwoSampleMR/gwas.mrcieu.ac.uk). Reassuringly, the strongest associations involved blood cell counts and traits and lymphoid and hematopoietic cancers, but we also uncovered new associations such as with malignant skin cancers. We have incorporated these analyses in new Supplementary Tables 40 and 41 and updated the Results section (page 8) under “Mendelian randomization (MR) to uncover the causes and consequences of CH” and the corresponding entry for MR analysis in the Methods section (page 15).

5. Regarding the prioritized functional genes, enrichment analysis with target drugs is interesting. Is there enrichment with the targets of the diseases with altered comorbidity with CH (e.g., heart failure and neoplasm)?

We have now analyzed the prioritized functional genes (*SMC4*, *ENPP6*, *TERT*, *CD164*, *ATM*, *PARP1*, *TCL1A*, *SETBP1*, and *TMEM209*) in the context of drug target information from the Open Targets Platform (platform.opentargets.org; now reported in Supplementary Table 31) and the canSAR (cansarblack.icr.ac.uk; now in Supplementary Table 32) integrated knowledge-base that brings together multidisciplinary data to provide useful predictions for drug discovery. PARP inhibitors are currently approved for a range of solid tumors and in trials for other malignancies while inhibitors of TERT are in trials for myelofibrosis and myelodysplastic syndrome (Supplementary Table 31). SMC4 and ATM have high chemistry-based “druggability” scores while ENPP6 and CD164 have antibody-based “druggability”. We have now added new text linked to the aforementioned new Supplementary Tables under the Results and Methods sections under “Functional target gene prioritization at CH risk loci” (pages 6-7) and under “Fine-mapping and target gene prioritization” (page 14), respectively. We also note, in relation the Reviewer’s point, that under the Discussion section on page 8 we discuss that “New CH risk loci included the *PARP1* coding variant rs1136410, where the G allele is protective for *DNMT3A*-CH and associated with reduced catalytic activity suggesting that this most common form of CH may be vulnerable to PARP inhibition, in keeping with the observed synergy between PARP and DNMT inhibitors.”

6. How about the GWAS results of the non-European UK biobank participants?

This is a very important point and we thank the reviewer for raising it. After removal of genetically inferred European ancestry UK Biobank (UKB) participants and sample exclusions due to quality control we were left with 505 individuals with any form of CH and 11,893 controls of non-European ancestry. This non-European ancestry sub-cohort includes several ancestral groups and for any specific ancestry group and CH subtype we have an even smaller sample size (i.e., < 505 CH “cases”). Given this background we are grossly underpowered for common variant discovery analyses in multi-ancestry (non-European) UKB participants and do not undertake a full GWAS in this ancestrally diverse sub-cohort.

However, in the revised resubmission, we now evaluate the 7 lead variants identified for overall CH (the CH phenotype with maximum sample size in our study) in the multi-ancestry sub-cohort described above. We recognize that pooling such a diverse group and analyzing it as one is not ideal (we adjust our analyses for age, sex, exome sequence batch and 40 genetic ancestry principal components). Nevertheless, we find that 6/7 variants had directionally consistent effect size point estimates between the European and non-European ancestries and for 7/7 variants the non-European confidence interval included the European point estimate suggesting that these loci have consistent effects in both sub-cohorts (we provide a copy of the new Extended Data Fig. 4 displaying these results here for the Reviewer’s convenience). We include these new findings in the Results section under “Germline genetic loci associated with overall CH susceptibility” (page 4) with a corresponding Methods entry under “Germline genotype data processing and genome-wide association analyses” (page 12).

Reviewer #3:
Remarks to the Author:

In the manuscript “Genome-wide analysis of 200,453 individuals yields new insights into the causes and consequences of clonal hematopoiesis” Kar et al. use genetic epidemiology approaches to describe novel germline associations with CH and define a driver specificity for these associations that may explain phenotypic differences observed in DNMT3A and TET2 CH for example.

Overall the amount of analysis the authors have done in this work is commendable. However, there are several concerns regarding the manuscript focus and flow that need to be addressed. There are a very large number of observations reported, but they are not always related or well discussed. A large portion of the findings have been reported previously in other datasets. Some are novel, some p-values that are significant in prior datasets are not significant here, and some p-values are significant here that were not significant in prior datasets. I wonder whether the cleanest paper is to focus more on the germline predisposition findings, which are clearly better powered in this larger dataset compared to previous studies.

We thank Reviewer #3 for their positive comments and also for their valid criticism that our manuscript reports a large number of observations, including some that were reported before. As regards non-GWAS/MR data, our aims were to: i) reaffirm that the combination of the UKB exome data and our mutation calling approach robustly replicated previous findings and ii) to

use the statistical power afforded by the very large number of individuals we study to identify novel associations. Nevertheless, we do take on board our Reviewer's valid message and in response we have now abridged the phenotypic parts of our manuscript in clearer figures and condensed the relevant discussion in order to enhance novel findings (please see below).

1. The manuscript reads as multiple separable efforts that include the phenotype of CH, CH pathogenesis, germline associations. The bulk of the analysis focuses on CH pathogenesis and germline associations. However the authors begin the paper describing (on pages 2-3) phenotypic associations (prevalent and incident diseases) and it is extremely challenging for the reader to relate these findings with those of the germline associations. The discussion on how identified associations may be mechanistically linked to the outcomes described is minimal. Some of the associations observed have confidence intervals that approach 1 (For example the association of BMI has a modest p-value and 95% CI of 1.01-1.31) which call into question the real-world relevance of the statistical association. Though the experimental approach is observational, the work would be significantly strengthened by a discussion of potential mechanism that links important associations. Without this discussion, the volume of associations made in this manuscript without a clear focus does more to convolute understanding of CH consequences.

We thank the Reviewer for this comment that has helped us prepare a clearer resubmission with a sharper focus. To improve the flow of our manuscript, we have condensed the main Results text relative to Figures 1 and 2 to give only a concise overview of CH mutations in the 200k UK Biobank cohort, and now report only the most salient phenotypic associations as recommended by the Reviewer in the first three sub-headings under the Results sections on pages 2 and 3. Also, in terms of mechanisms, we now discuss on page 8 a recent preprint (Weinstock, et al. 2021; <https://doi.org/10.1101/2021.12.10.471810>) which suggests that *TET2* and *ASXL1* somatic mutations interact with a *TCL1A* promoter germline variant found to be associated with clonal expansion rate. While *TCL1A* is not expressed in normal or *DNMT3A*-mutated hematopoietic stem cells, the authors provide evidence that the locus becomes susceptible to activation in the presence of *TET2* or *ASXL1* mutations depending on the allele at the promoter variant, leading to faster clonal expansion. This type of interaction may operate for *CD164* and other CH risk loci, or alternative models of interaction between the germline and somatic genome may exist.

Finally, while we acknowledge that some of the Mendelian randomization (MR) associations observed have confidence intervals that approach 1 (for example, the BMI MR result quoted in this comment), we note that this does not necessarily mean that the associations have no real-world relevance. The clinical translation of MR effect sizes is an emerging area and there are notable examples of modest effect sizes marking clinically significant correlations. One such example is the modest association between LDL cholesterol-lowering alleles of *HMGCR* and coronary artery disease risk (OR = 0.97, 95% CI = 0.95-0.98 based on MR analysis in 60,801 CAD cases and 123,504 controls from the CARDIoGRAMplusC4D consortium; Lotta, et al. 2016; <https://pubmed.ncbi.nlm.nih.gov/27701660/>), which contrasts with the strong benefit of *HMGCR* inhibitors such as statins in preventing coronary artery disease.

2. Similarly, the figures do not highlight the novel findings or relate well to each other. Figure 1 is very consistent with prior reported findings with minimal novelty. Figure 2 has many findings that have been reported before and some novel ones, but they are buried. Figure 4 is the most clear figure in terms of what is known and novel.

We again thank the Reviewer for this comment that has also helped us focus our resubmission. We agree that Figure 1 is consistent with prior reports, but we included it because we believe that it is important to provide a brief overview of CH in order to demonstrate its proper identification and present its characterization in this UK Biobank exome cohort. In addition, to the best of our knowledge, the sex-associated differences highlighted in the distribution of CH mutations in Figure 1 is novel for CH (although some have been reported for myeloid neoplasms). **Following the Reviewer's suggestions, we have revised Figures 1 and 2 and Extended Data Figs. 2 and 3 to provide clearer evidence of the novelty of the observed associations and substantially reduced the main text corresponding to Figures 1 and 2 under the Results sections (pages 2-3).**

3. The significance of the ATAC-Seq data and the implications of these findings are not entirely clear to me.

We apologize for not making this clearer. ATAC-Seq identifies genomic regions of "open chromatin" that are transcriptionally active and previous work has established that both genome-wide significant ($P < 5 \times 10^{-8}$) disease- or trait-associated variants (Corces, et al. 2016; <https://pubmed.ncbi.nlm.nih.gov/27526324/>) and the heritability of diseases and traits (the whole-genome genetic association signal including associated genetic variation that does not reach the $P < 5 \times 10^{-8}$ threshold; Finucane, et al. 2018; <https://pubmed.ncbi.nlm.nih.gov/29632380/>) are enriched in open chromatin regions in the corresponding disease/trait-relevant cell types. We therefore used ATAC-Seq data in two places in the manuscript. **First**, to show for the first time that the heritability for CH is enriched in open chromatin regions identified by ATAC-Seq in hematopoietic stem and progenitor cells/lineages (Fig. 3). This affirms the robustness of our GWAS data and links, in particular, open chromatin regions in hematopoietic stem cells, common lymphoid and myeloid progenitors, and multipotent and erythroid progenitors to our GWAS data. **Second**, we overlap fine-mapped variants at the genome-wide significant CH risk loci reported in our manuscript with the hematopoietic cell ATAC-Seq peaks and explore the correlation between these peaks and expression (RNA-Seq) profiles of nearby genes from the same hematopoietic cells to connect the fine-mapped variants to their most likely target genes, implicating *SMC4*, *ATM*, *PARP1*, and *TCL1A* (Fig. 5c) via this application of the ATAC-Seq data which leverages a pipeline that was previously described in Ulirsch, et al. 2019; <https://pubmed.ncbi.nlm.nih.gov/30858613/> and Bao et al. 2020; <https://pubmed.ncbi.nlm.nih.gov/33057200/>. **We have now expanded on the significance and implications of the ATAC-Seq data in the Results section under "Heritability and cell type-specific enrichment of polygenic susceptibility to CH" (page 3) and under the Discussion (page 8).**

4. Are p-values adjusted for multiple hypotheses testing? This is particularly important in assessing the relevance of mild-modest effect estimates with mild-modest p-values.

Associations between (1) CH and traits/diseases present at the time of blood sampling in the UK Biobank (UKB) and (2) CH and incident disease were adjusted for multiple hypotheses testing using the false discovery rate (FDR). **We apologize for the lack of clarity on this in the original submission and have now clarified this in the Methods section under "Trait selection**

and modelling for the conventional observational multivariable regression analyses” (page 11) and in the caption for revised Extended Data Fig. 3a, and provide the FDRs in Supplementary Tables 6 to 8 and 10 to 13.

For the Mendelian randomization (MR) analyses, in the form presented in the original submission, our focus was on appraising the roles of a targeted set of potential causes and causal consequences of CH and in this specific setting (which is distinct from large-scale hypothesis-free data exploration) we chose to not adjust for multiple comparisons. We believe that in this setting, to quote Kenneth Rothman’s classic and highly cited 1990 paper (<https://pubmed.ncbi.nlm.nih.gov/2081237>), “a policy of not making adjustments for multiple comparisons is preferable because it will lead to fewer errors of interpretation when the data under evaluation are not random...”. One could have presented each of our MR results in separate standalone publications without incurring any penalty for multiple comparisons testing but we believe that our presentation of these targeted MR results in a single paper will move the field of CH research forward much more substantially.

Finally, it is also worth noting here that Reviewer 2 (question 4) suggested that we undertake a Mendelian randomization phenome-wide association study (MR-PheWAS) using genetic data on outcomes from the UK Biobank. Therefore, in addition to the targeted MR analyses originally presented, we have now also evaluated 1,484 diseases and traits as potential consequences of CH and find, amongst several such hematological examples, that genetic predisposition to overall CH is associated with cancers of primary lymphoid and hematopoietic tissue with a P -value of 9.4×10^{-3} and FDR of 0.34 (Supplementary Tables 40 and 41). It is widely accepted that CH is causally associated with cancers of primary lymphoid and hematopoietic tissue but given this MR P -value, applying multiple comparisons adjustment to the MR analyses would have led us to discard this association underscoring the risk of a type II error as highlighted by Rothman. Nonetheless, we do provide FDRs in the two new MR-PheWAS supplementary tables given that the MR-PheWAS is a hypothesis-free approach.

5. With regards to the phenotype data, in the section “associations between CH and incident disease” on page 3, the authors report that CH is not associated with ischemic cardiovascular disease in the UKB. They do not provide any discussion of this novel finding – are there differences in cohort composition for example that might explain this?

We are grateful to the Reviewer for this critically important question. A very similar point is also raised by Reviewer #1 (question 2) and we provide below our consolidated response to this point for both our Reviewers.

We now follow the suggestion of Reviewers #1 and #3 to explore the evidence for confounding by age and smoking further and examine the association between CH and coronary artery disease (CAD)/stroke with and without age and/or smoking as covariates. We present results for four models in the figure copied below – univariate (including CH as the only predictor), bivariate with CH + age, bivariate with CH + smoking, and trivariate with CH + age + smoking as predictors. We apply these models to myocardial infarction (MI), CAD, ischemic stroke, overall stroke, heart failure (HF) and atrial fibrillation (AF) as individual outcomes, as well as to a composite of cardiovascular outcomes. As evident in the forest plots below, we find that the associations between CH and CAD/MI and CH and stroke/ischemic stroke attenuate to the null once age is included in the models. While there is also substantial attenuation of the associations between CH and HF and CH and AF in models that include age, the HF and AF associations continue to remain significant. The impact of smoking on the associations in the models is relatively small compared to the profound impact of age as a confounder, although in every case the trivariate models that include both age and smoking are the ones closest to or at the null. We now present these new analyses in Extended Data Fig. 3c and

Supplementary Table 12 and comment on this under “Associations between CH and incident disease” in the Results section (page 3) and in the Discussion section (page 9).

We also note that there are other recent publications that support our finding of a lack of an association in the UK Biobank (UKB). Notably, Bhattacharya, et al. (2021; <https://pubmed.ncbi.nlm.nih.gov/34743536/>), published after we first submitted our manuscript, reported equivalent findings for the association between stroke and CH in the UKB. For convenience, we have copied below the relevant figure (Supplementary Fig. S12F) from Bhattacharya, et al., which demonstrates the lack of an association between CH and ischemic stroke amongst ~45,000 UKB participants (a subset of the ~200k analyzed in our study). The confidence interval for the association continues to include the null even after inclusion of multiple other studies, including several studies that are highly enriched for stroke outcomes.

Figure 1. Forest plot of meta-analyzed hazard ratio for the association between clonal hematopoiesis of indeterminate potential (CHIP) and stroke.
 Cox proportional hazards models were fitted, adjusted for age, sex, and the first 10 principal components of genetic ancestry. Here, forest plots are used to show the hazard ratio (HR), 95% CI, and numerical events for each study. **A.** CHIP was associated with a meta-analyzed HR of 1.14 (95% CI, 1.03–1.27) for all incident stroke. **B.** CHIP was associated with a meta-analyzed HR of 1.11 (95% CI, 0.98–1.25) for all incident ischemic stroke. **C.** CHIP was associated with a meta-analyzed HR of 1.24 (95% CI, 1.01–1.51) for all incident hemorrhagic stroke. ARIC indicates Atherosclerosis Risk in Communities; CHS, Cardiovascular Health Study; FHS, Framingham Heart Study; JHS, Jackson Heart Study; MESA, Multi-Ethnic Study of Atherosclerosis; MGBB, Mass General Brigham Biobank; UKBB, United Kingdom Biobank; and WHI, Women's Health Initiative.

It is noteworthy that many of the cohorts that have reported the association are enriched in participants at high cardiovascular risk (e.g., Bioimage, ATVB, and PROMIS in Jaiswal, et al. 2017). One plausible mechanistic explanation for the lack of an association in the UKB is that the UKB cohort, in contrast, is relatively younger and healthier and may therefore potentially have lower rates of epigenetic aging. Recently published findings, also from the Jaiswal group (Nachun, et al. 2021; <https://pubmed.ncbi.nlm.nih.gov/34050697/>), suggest that CH was associated with CAD/stroke only on a background of epigenetic aging, whereas the association was not seen in those without evidence of epigenetic age acceleration. **Based on our Reviewer #3's suggestion to provide an expanded discussion on mechanisms we include the above under the Discussion section (page 9).**

6. On page 4, the authors describe associations of germline *TERT* variants and long leukocyte telomere length with CH. Within the discussion, this finding should be reconciled this with recent data from Reilley et al Blood 2021 (doi: 10.1182/blood.2021011075) which identifies germline variants and shortened telomeres as being associated with myelodysplastic syndrome.

We thank the reviewer for pointing us to this interesting manuscript that identifies deleterious germline *TERT* rare variants and shortened telomeres in MDS patients without a clinical diagnosis of a telomere biology disorder. Intriguingly, the authors show that the distribution of somatic mutations in these patients differs from that seen in conventional MDS (Supplemental Figure 4 in their paper). Most notably, these cases displayed a striking paucity of somatic mutations in *DNMT3A* (2/41 cases) and *TET2* (3/41 cases), the two most common drivers of CH. As mutations in *DNMT3A* and/or *TET2* are seen in ~50% of MDS (i.e., a prevalence similar to that seen in CH), these findings suggest that evolutionary paths may differ between cases with long versus short telomeres. **We now discuss the Reilley et al Blood 2021 paper on page 9 of our manuscript.**

7. On page 5 the authors discuss the hematological association of autosomal mCA with CH. The relationship between CH and autosomal mCA in both myeloid and lymphoid CH has recently been described using data from the UK Biobank. The authors should reference this in their discussion. <https://doi.org/10.1038/s41591-021-01521-4>

Thank you for pointing us to this recent publication. We have now cited this in our discussion of the relevant findings under “Genetic relationship between hematological chromosomal mosaicism and CH due to gene mutation” in the Results section (pages 5-6) where we also connect this recent publication with our result that the heritability of overall CH is enriched in open chromatin regions (profiled by ATAC-Seq) in common lymphoid and myeloid progenitors, among other cell types (Fig. 3b and in response to the Reviewer’s own question above on ATAC-Seq).

Minor suggestions:

- In abstract on page 1, line 36 and on page 8, line 363 – would change text to “CH from 4 to 14” for uniformity

We have changed the text in both places.

- The text within Figure 4 is difficult to read within the boxes given the chosen colors

Colors have been changed and we hope that the updated colors make the text more readable.

- In extended figure 3b are myeloid neoplasms a composite of several ICD10 codes or derived from codes shown in figure 3a. This needs to be specified in the methods. “Thrombocytopenia unspecified” should not be used as this is a basket code for many different causes of thrombocytopenia, most non-malignant.

Myeloid malignancies are a composite of several codes, including self-reported cancer, self-reported illness, and ICD-10 and ICD-9 codes. Definitions are provided in the revised Supplementary Table 9. Regarding the category of “Thrombocytopenia unspecified”, this phenotype has not been manually selected, but appeared as significant after a phenome-wide analysis of 11,787 ICD-10 codes. We have updated the figure for that analysis to clarify this (please see revised Extended Data Fig. 3a) and also describe this analysis in greater detail under “Trait selection and modelling for the conventional observational multivariable regression analyses” in the Methods section (page 11).

Decision Letter, first revision:

Our ref: NG-A58701R

4th Mar 2022

Dear Dr. Vassiliou,

Thank you for submitting your revised manuscript "Genome-wide analyses of 200,453 individuals yields new insights into the causes and consequences of clonal hematopoiesis" (NG-A58701R). It has now been seen by the original referees and their comments are below. The reviewers find that the paper has improved in revision, and therefore we'll be happy in principle to publish it in Nature Genetics, pending minor revisions to satisfy our editorial and formatting guidelines.

Sincerely,

Safia Danovi
Editor
Nature Genetics

Reviewer #1 (Remarks to the Author):

The authors have thoroughly addressed my comments. The responses regarding replication and CAD/stroke associations were very thorough. In particular, I appreciated the reference to Bhattacharya 2021 which indeed supports the findings of this paper and is a relevant citation.

Reviewer #2 (Remarks to the Author):

Authors fully addressed the reviewer's comments. This reviewer has no further comments.

Reviewer #3 (Remarks to the Author):

The authors have provided a thorough response to the reviewer comments.

Final Decision Letter:

In reply please quote: NG-A58701R1 Vassiliou

6th Jun 2022

Dear Dr. Vassiliou,

I am delighted to say that your manuscript "Genome-wide analyses of 200,453 individuals yield new insights into the causes and consequences of clonal hematopoiesis" has been accepted for publication in an upcoming issue of Nature Genetics.

Your paper will be published online after we receive your corrections and will appear in print in the next available issue. You can find out your date of online publication by contacting the Nature Press Office (press@nature.com) after sending your e-proof corrections. Now is the time to inform your Public Relations or Press Office about your paper, as they might be interested in promoting its publication. This will allow them time to prepare an accurate and satisfactory press release. Include your manuscript tracking number (NG-A58701R1) and the name of the journal, which they will need when they contact our Press Office.

Acceptance is conditional on the data in the manuscript not being published elsewhere, or announced in the print or electronic media, until the embargo/publication date. These restrictions are not

intended to deter you from presenting your data at academic meetings and conferences, but any enquiries from the media about papers not yet scheduled for publication should be referred to us.

Please note that *Nature Genetics* is a Transformative Journal (TJ). Authors may publish their research with us through the traditional subscription access route or make their paper immediately open access through payment of an article-processing charge (APC). Authors will not be required to make a final decision about access to their article until it has been accepted. [Find out more about Transformative Journals](https://www.springernature.com/gp/open-research/transformative-journals)

Authors may need to take specific actions to achieve [compliance with funder and institutional open access mandates](https://www.springernature.com/gp/open-research/funding/policy-compliance-faqs). If your research is supported by a funder that requires immediate open access (e.g. according to [Plan S principles](https://www.springernature.com/gp/open-research/plan-s-compliance)) then you should select the gold OA route, and we will direct you to the compliant route where possible. For authors selecting the subscription publication route, the journal's standard licensing terms will need to be accepted, including [self-archiving and license to publish](https://www.nature.com/nature-portfolio/editorial-policies/self-archiving-and-license-to-publish). Those licensing terms will supersede any other terms that the author or any third party may assert apply to any version of the manuscript.

Please note that Nature Portfolio offers an immediate open access option only for papers that were first submitted after 1 January, 2021.

If you have not already done so, we invite you to upload the step-by-step protocols used in this manuscript to the Protocols Exchange, part of our on-line web resource, natureprotocols.com. If you complete the upload by the time you receive your manuscript proofs, we can insert links in your article that lead directly to the protocol details. Your protocol will be made freely available upon publication of your paper. By participating in natureprotocols.com, you are enabling researchers to more readily reproduce or adapt the methodology you use. [Natureprotocols.com](http://natureprotocols.com) is fully searchable, providing your protocols and paper with increased utility and visibility. Please submit your protocol to <https://protocolexchange.researchsquare.com/>. After entering your nature.com username and password you will need to enter your manuscript number (NG-A58701R1). Further information can be found at <https://www.nature.com/nature-portfolio/editorial-policies/reporting-standards#protocols>

Sincerely,

Safia Danovi
Editor
Nature Genetics